# Magnetic nanostickers for active control of interface-enhanced selective bioadhesion

Changshun Hou [1] ✉, Junjia Guo [2], Bonan Sun [1], Kai Fung Chan [2,3], Xin Song [1,4] & Li Zhang [1,2] ✉

Natural biological tissues exhibit different mechanical and surface properties. These disparate features make their connections with engineering materials quite difficult due to the lack of universal methods for tuning the interfacial bonding over a wide range. However, the precise control of interfacial properties, including modulus and adhesion on diverse biological tissues, requires overcoming multiple inherent and external barriers. Here we propose an interface-enhanced strategy by spatial and temporal anchoring of magnetic nanostickers for controlled bioadhesive properties. Fully exploiting the interactions from nanostickers by remote control enables the attached patch to achieve extremely high adhesion energy ($\sim 1250\,\mathrm{J\,m^{-2}}$) and interfacial fatigue resistance with a threshold of $\sim 50\,\mathrm{J\,m^{-2}}$, at a very low area density of nanostickers ($4\,\mu\mathrm{g/mm^2}$). The controlled interfacial properties as well as space and time for anchoring, lead to comprehensively tunable bioadhesion on diverse tissues such as skin, intestine, liver, and kidney, which are strongly desired in biomedical applications. Integration with fragile tissues in female Sprague-Dawley rats for 10 days further demonstrates that the anchored biointerface can adapt to the in vivo environment and promote postoperative recovery. The biointerface bridged by intelligent nanostickers prompts the methodology for bioadhesion towards controllable orientation.

Aqueous nanoparticle solutions for bridging tissues with polymers has shown potential in clinical practice[1,2], but they rely on uncontrollable diffusion to connect with the tissues or polymers, resulting in uncertain bioadhesive properties. In contrast, the precise and controlled bioadhesion is highly desired in in vivo applications because it can facilitate the precise medical treatment. A rational approach to addressing these issues may involve control systems to steer the anchoring agents with sufficient driving force. The emergent high-power ultrasound has innovated the application of hydrogels with high adhesion energy and interfacial fatigue threshold on pig skin by pressure-gradient propulsion of the anchoring primers[3]. However, contact ultrasound probes may discomfort the body and are

unsuitable for activating adhesion on fragile parts (e.g., diseased regions) and deep tissues (Supplementary Table 1)[4–6]. Current interfacial bonding technologies have not yet demonstrated that bioadhesion can be controlled comprehensively according to our demands.

Setting up robust biointerfaces between dynamic wet biological tissues and hydrogels holds a significant promise for medical electronics and tissue regeneration[7–11]. Different strategies using adhesive bonds and topological connections to form tissue-hydrogel hybrids serve as alternatives or adjuncts for conventional sutures, staples, clips, and commercial bio-glues[12–16]. Although reliable biointerfaces can secure hydrogels on targeted tissues to generate effective adhesion, they are difficult to form and challenging to control in terms of

[1]Department of Mechanical and Automation Engineering, The Chinese University of Hong Kong, Shatin N.T. Hong Kong, China. [2]Department of Biomedical Engineering, The Chinese University of Hong Kong, Shatin N.T. Hong Kong, China. [3]Li Ka Shing Institute of Health Sciences, The Chinese University of Hong Kong, Shatin N.T. Hong Kong, China. [4]Department of Biomedical Engineering, City University of Hong Kong, Kowloon Hong Kong, China.
✉e-mail: changshou2@cuhk.edu.hk; lizhang@mae.cuhk.edu.hk

interfacial properties and space and time for anchoring. For example, chemical anchoring using carbodiimide chemistry suffers from spatial and temporal uncontrollability and poor interfacial fatigue resistance[17,18]. Furthermore, the limited availability of functional groups and external barriers such as biofluids and stratum corneum severely impede the anchoring agents from forming an interpenetrating network with tissues[19,20].

Micro/nanomaterials powered by different exogenous energy sources such as light[21], chemicals[22], and magnetism[23], have shown high controllability in their mechanical motion within narrow and confined lumens[24], which distinguishes them from the behaviors of passive micro/nanomaterials. Compared with other types of actuation, magnetic field have remarkable advantages including harmlessness, programmability, transparency, and deep penetration, allowing remote and precise steering of micro/nanomaterials by applying appropriate force and torque[25,26]. Although an individual micro/nanomaterial provides minimal bonding energy with tissues (nanoparticle-tissue membranes: −4.4 to +5.1 kJ mol$^{-1}$)[27], the collective interactions particularly for the magnetic dipole-dipole interactions between the programmed micro/nanomaterials could synergistically augment their cohesive force and thus improve the interfacial adhesion energy[28]. The repertoire of micro/nanomaterials lies in bridging tissues and applied hydrogels through a remote magnetic field. Guiding the magnetic micro/nanomaterials to anchor on tissues requires strong propulsion to pass through the external barriers[29], and once settled, these micro/nanomaterials can immobilize the hydrogels. For secure immobilization and high controllability on tissues, the prepared micro/nanomaterials exhibit stable superparamagnetic properties with high magnetic saturation. In the meantime, surface functionalization of the micro/nanomaterials can introduce abundant intermolecular interactions such as hydrogen bonding and Coulomb force with the hydrogels for interfacial gelation.

In this work, the magnetic control bioadhesion is implemented by a robust hydrogel patch and cationic biopolymer-coated superparamagnetic nanostickers (Supplementary Fig. 1). This strategy for enhancing tissue-hydrogel interfaces not only relies on conventional energy dissipation by a soft hydrogel layer, but also incorporates hard nanostickers around the tissue-hydrogel interface to enhance resistance to interfacial failure. The robust mechanical properties of the hydrogel patch can also prevent the scission of the interfacial hydrogel layer and unfavorable swelling in biofluids. Taking advantages of the integrated energy dissipation and interfacial enhancement, a minimal area density of nanostickers (4 μg/mm²) can realize an ultrahigh adhesion energy around 1250 J m$^{-2}$ and interfacial fatigue threshold of ~50 J m$^{-2}$. Adhesive properties can be further precisely controlled by magnetic parameters, showing tunable ability for different biomedical applications.

## Results
### Precise control of bioadhesion
The process for interfacial enhancement of bioadhesion includes the dispersion and wireless actuation of nanostickers, followed by simply attaching a hydrogel patch (Fig. 1A). The well-dispersed nanostickers facilitate manipulation and subsequent anchorage, which can be used by spraying, dropping, and brushing. A gradient-based rotating magnetic field beneath the tissues can rotate and propel the nanostickers, providing controlled adhesive properties through active repulsion of surrounding barriers and penetration into the tissue surface. Multiple interactions from the nanostickers, including strong magnetic attraction and crosslinks with the robust patch, ensure together a stable

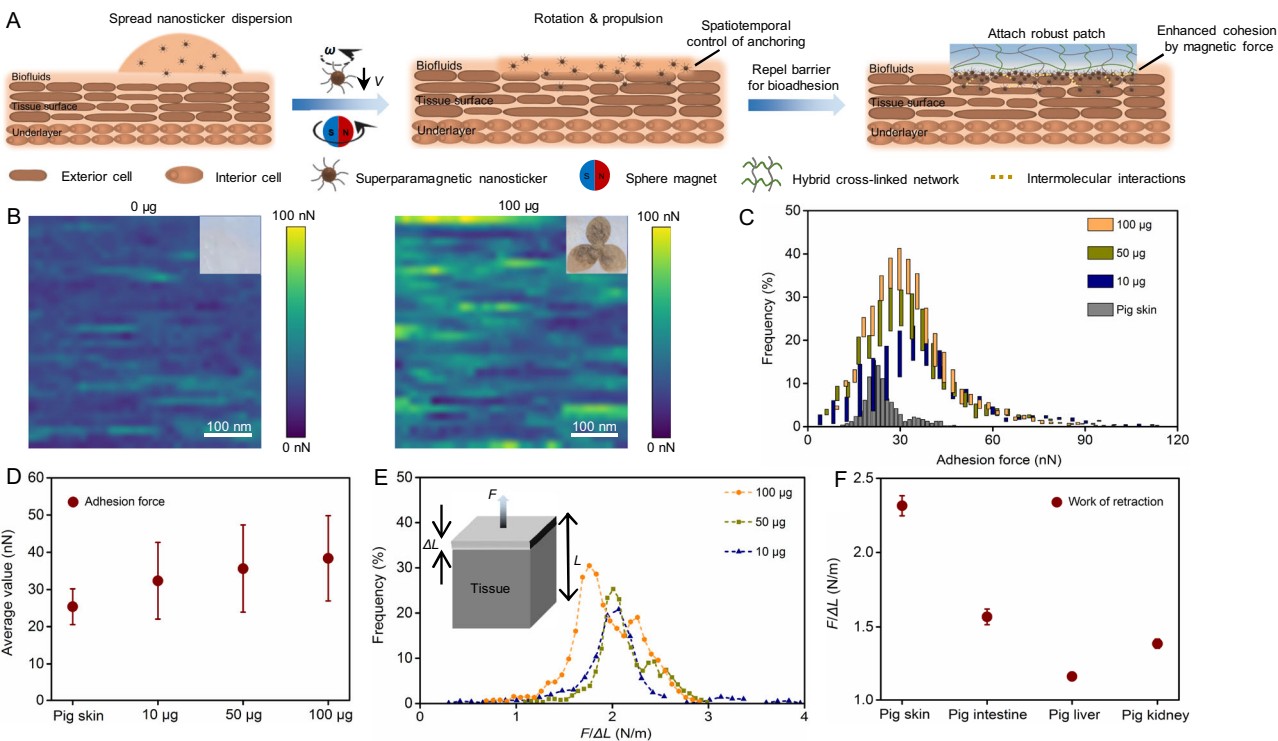

**Fig. 1 | Precise control of magnetic nanostickers for enhanced interfacial bioadhesion. A** Schematic illustrating the magnetic control of nanostickers to bridge tissues with hydrogel patches. **B** Adhesion force mapping compares the adhesion force distribution between the nanostickers-anchored surface and the pig skin surface. Inset shows real images of nanostickers' anchorage. **C** Adhesion force distribution of the anchored skin with different quantities of nanostickers.

**D** Average adhesion force generated by different quantities of nanostickers. All values are presented as mean ± SD for $n = 3$ independent experiments. **E** Work of retraction produced by different quantities of nanostickers. **F** Average work of retraction measured on diverse tissues (50 μg of nanostickers). All values are presented as mean ± SD for $n = 3$ independent experiments.

tissue-patch interface. In detail, the strong magnetic attraction among the anchored nanostickers can generate high cohesive force to enhance the interface between the tissue and patch, and the cationic polymers on the surface of nanostickers further provide cross-linking sites with the attached patch through typical charge and hydrogen bonding interactions[30]. These interactions together make the adhesive interface robust with high fatigue resistance. To verify the universality and feasibility of magnetic control method in bioadhesion, nanostickers are prepared with various cationic polymers (chitosan, polyethyleneimine, and gelatin) for standard 180° peel tests (Supplementary Fig. 2). Compared to the soft polyethyleneimine and gelatin with limited amino groups, the molecular structure of chitosan is rigid and has abundant amino groups[15], making the $Fe_3O_4$@chitosan have the highest adhesion energy. Despite differences in charge density and structure of the coated polymer, all prepared nanostickers can distinctly increase their adhesion energy on robust pig skin. We also investigate the adhesion effects of different dual-cross-linked hydrogel patches (PAAM-Alg, PAAM-Agar, and PNIPAM-Alg) (Supplementary Fig. 3A), because the hybrid network usually can provide exceptional mechanical properties compared to single networks[31,32]. The adhesion energy increases with improved mechanical robustness of hydrogel patches, suggesting that the interfacial bridging by nanostickers is robust enough to contend against the mechanical deformation of patches (Supplementary Fig. 3B)[33]. These results indicate that a variety of nanostickers and hydrogel patches can be applied with magnetic control method. To study the interfacial regulation by nanostickers upon peeling, we choose $Fe_3O_4$@chitosan and polyacrylamide-alginate (PAAm-Alg) for following demonstration. The robust PAAm-Alg patch (fracture stress ~368 KPa) promotes failure occurs at the tissue-hydrogel interface instead of the bulk. The prepared $Fe_3O_4$@chitosan nanostickers are optimal for interfacial bridging, exhibiting a small diameter averaging around 10 nm, positive surface potential at +19.6 mV, and stable superparamagnetic saturation at 58 A·m²·kg⁻¹ (Supplementary Figs. 4–7). Magnetic dipole-dipole interactions are verified when the scattered $Fe_3O_4$@chitosan nanostickers present strong magnetic attraction (Supplementary Fig. 8).

Controlling the nanostickers with different spatial patterns on tissues can be realized by direct magnetic steering or mask-assisted anchoring. The pattern shape formed by the direct magnetic control is highly determined by the contact shape of nanosticker dispersion on tissues (Supplementary Fig. 9A, B). The mask-assisted anchoring facilitates the formation of complex patterns with various resolutions. For example, clover-shaped patterns with small sizes of 10 mm, 15 mm, and 20 mm are formed by masking the skin and wirelessly steering the nanostickers at different area densities (Supplementary Fig. 9C, D). With the quantity of nanostickers applied at 10 μg, 50 μg, and 100 μg, the area density increases accordingly to $0.37 \, \mu g/mm^2$, $0.82 \, \mu g/mm^2$, and $0.92 \, \mu g/mm^2$, respectively. The bonding state of nanostickers is examined by scanning electron microscope (SEM), revealing that the compact nanostickers are highly cohesive and firmly anchored on the pig skin (Supplementary Fig. 10). Water blasting test further substantiates that the nanostickers are tightly anchored on the skin surface, which can withstand high impact without visible damage (Supplementary Fig. 11). Atomic force microscope (AFM) is then used to measure the adhesion force generated at different area densities. Compared to the pig skin surface, adhesion force mapping shows a remarkable elevation of the adhesion force on the nanostickers-anchored surface (Fig. 1B). The average adhesion force of the pig skin surface is 25.35 nN, which increases to 32.33 nN, 35.61 nN, and 38.37 nN by raising the area density of anchored nanostickers from $0.37 \, \mu g/mm^2$ (10 μg) to $0.92 \, \mu g/mm^2$ (100 μg) (Fig. 1C, D). Moreover, the retraction force and distance are accurately recorded by the AFM tip, revealing the work of retraction ($F/\Delta L$) from the anchored nanostickers (Fig. 1E). The work of retraction is elevated from 2.08 N/m to 2.32 N/m when the area density of nanostickers increases from $0.37 \, \mu g/mm^2$ to $0.82 \, \mu g/$

mm². However, further increasing the area density to $0.92 \, \mu g/mm^2$ decreases the work of retraction to 2.04 N/m. This reduction should be caused by the excessive nanostickers that cannot be adequately anchored, as evidenced by the presentation of two peaks in the retraction curve. The anchored nanostickers also improve the work of retraction measured on diverse tissues, indicating the magnetic control method has broad applicability (Fig. 1F).

## Controlled bioadhesive properties

To elucidate the controllable bioadhesion between the skin and patch, a series of mechanical tests are performed. Lap-shear tests reveal that the shear strength of the anchored skin-patch hybrid can reach 187 KPa, which is more than 5 times of the shear strength tested without nanostickers (Fig. 2A). The anchored skin-patch interface unexpectedly shows that the interfacial strength is even stronger than the mechanical strength of the robust patch, providing direct evidence for interfacial enhancement between the skin and patch (Supplementary Fig. 12 and Supplementary Movie 1). The shear strength of the anchored skin-patch hybrid improves when the quantity of nanostickers increases from 0 mg to 2 mg (area density ranges from 0 μg/mm² to 4 μg/mm²). When the quantity of nanostickers exceeds this point to 3 mg (area density: 6 μg/mm²), the shear strength reduces to 74 KPa (Fig. 2B). This result suggests that the area density of nanostickers approaches saturation, for which the excessive nanostickers can impair the interfacial bridging with the skin and patch. Afterwards, peel tests demonstrate that the anchored skin-patch hybrid can achieve a high adhesion energy at 1250 J m⁻² with an average area density of nanostickers at 4 μg/mm², which is elevated for 90 times when compared to the adhesion energy measured without nanostickers (Fig. 2C, Supplementary Fig. 13, and Supplementary Movie 2). The area densities of nanostickers for optimized adhesion energy investigated by the AFM and peel tests show some variation since they are different testing methods. The quantity of nanostickers also significantly influence the adhesion energy (Fig. 2D), for which the highest adhesion energy is obtained when 2 mg of nanostickers is immobilized within an area of 2 cm × 2.5 cm ($W \times L$). This trend correlates well with the previous lap-shear tests. Taking advantage of the gradient-based rotating magnetic control for immobilization, we enable minimal nanostickers to obtain a high adhesion energy. The adhesion energy generated from per milligram of nanostickers is ~625 J m⁻² mg⁻¹ when the area density reaches 4 μg/mm² (Supplementary Table 2). The ultralow dosage of magnetic nanostickers, coupled with their adhesive performance, far outperforms previous nanoparticle-based adhesives (Fig. 2E)[3,34–36].

Cyclic peel tests are then performed to explore the fatigue-resistant adhesive properties (Supplementary Fig. 14). The cyclic energy release rate $G$, defined as $F_c/W$ ($F$ is the given peel force that less than the steady-state peel force; $W$ is the width of patch), is applied on the attached patch for $N$ cycles; and the interfacial crack propagation rate $d_c/d_N$ is measured from the interfacial crack extension $c$ and cycle number $N$. Different energy release rates are investigated to generate the plot of $d_c/d_N$ along with $G$, in which the interfacial fatigue threshold $\tau_O$ can be calculated by straightly extending the plot to the intercept of the $G$ axis[37,38]. The nanostickers elevate $\tau_O$ from ~2 J m⁻² to ~50 J m⁻², suggesting that the fatigue resistance of magnetic control bioadhesion is significantly better than the typical covalent bonding around 25 J m⁻² (Fig. 2F)[18]. Different from weak interactions of usual physical linkages such as the topological connection (Supplementary Table 1), the enhanced interface evidenced by these results is even stronger than the interfacial interactions of chemical linkages.

## Anchoring mechanism of nanostickers

To investigate the anchoring mechanism of nanostickers by the magnetic control method, we simulate the magnetic field used in the mechanical tests. A sphere permanent magnet with a 50 mm diameter

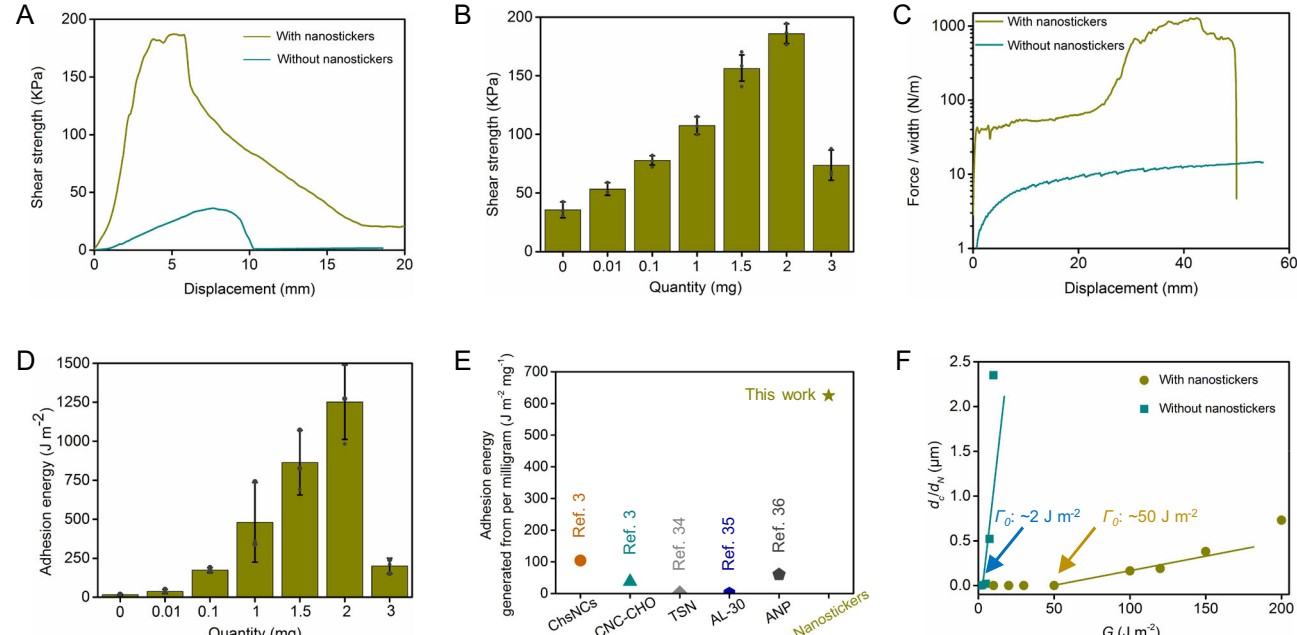

**Fig. 2 | Magnetic control of nanostickers for tunable adhesive properties.**
**A** Representative curves of different skin-patch hybrids by lap-shear tests. **B** Shear strength of the skin-patch hybrid anchored by different quantities of nanostickers. Data are presented as mean ± SD; $n = 3$ independent samples. **C** Representative curves of the skin-patch hybrids with and without nanostickers by 180° peel tests.

**D** Adhesion energy varies with different quantities of nanostickers. Data are presented as mean ± SD; $n = 3$ independent samples. **E** Comparison of the adhesive performance of nanostickers with other representative nanoparticle-based adhesives (ref. denotes as reference). **F** Interfacial crack propagation rate ($d_c/d_N$) versus different energy release rates $G$ at an average area density of $4\,\mu g/mm^2$.

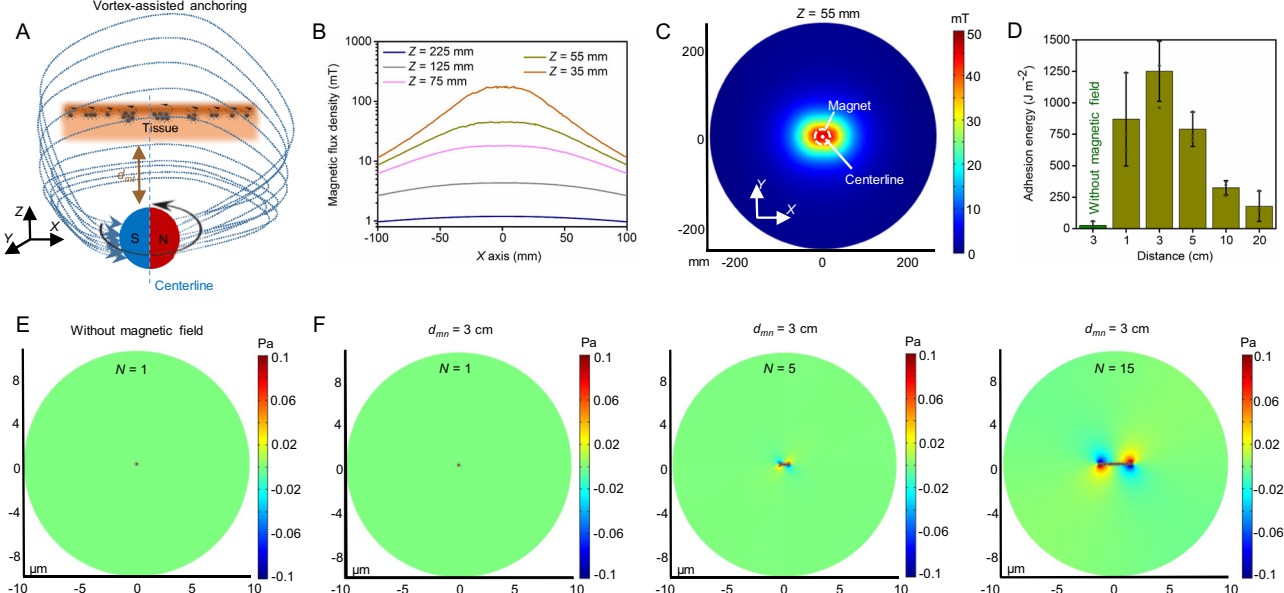

**Fig. 3 | Motion of nanostickers and anchoring mechanism. A** Schematic illustration of actuating the nanostickers with gradient-based rotating magnetic field. **B** Flux density distribution along the $X$-axis at different distance ($Z$). **C** A horizontal slice of flux density distribution at $Z = 55$ mm. **D** Adhesion energy varies with the

distance $d_{mt}$. Data are presented as mean ± SD; $n = 3$ independent samples. **E** Shear stress distribution of an individual nanosticker without magnetic control.
**F** Representative snapshots showing the increasing trend in shear stress as nanostickers assemble together.

is placed under the tissues, which can steer the nanostickers to rotate and propel into the tissue surface for anchoring (Fig. 3A). The magnetic field is parallel to the $XY$ plane and generated on the top of the magnet, and its strength can be easily controlled by adjusting the vertical distance between the top surface of the magnet and tissue surface $d_{mt}$. The magnetic field distribution shows that the magnetic flux density decreases with the increased distance of $Z$ ($Z = d_{mt} + r_m$, where $r_m$ is the radius of the magnet) and the in-plane distance to the centerline

(original point) (Fig. 3B). The practical distance in all tests is set as $Z = 55$ mm ($d_{mt} = 30$ mm) based on the magnetic flux density, and the simulation displays that the flux density changes a little (~0.4 mT at $d_{mt} = 30$ mm) for $X$-coordinate and $Y$-coordinate from −25 mm to 25 mm (Fig. 3C). Peel tests substantiate that the distance between the magnet and the skin surface can cause a remarkable variation of adhesion energy, and the skin-patch hybrid shows a very low adhesion energy at 26.8 J m⁻² without magnetic steering (Fig. 3D). In addition,

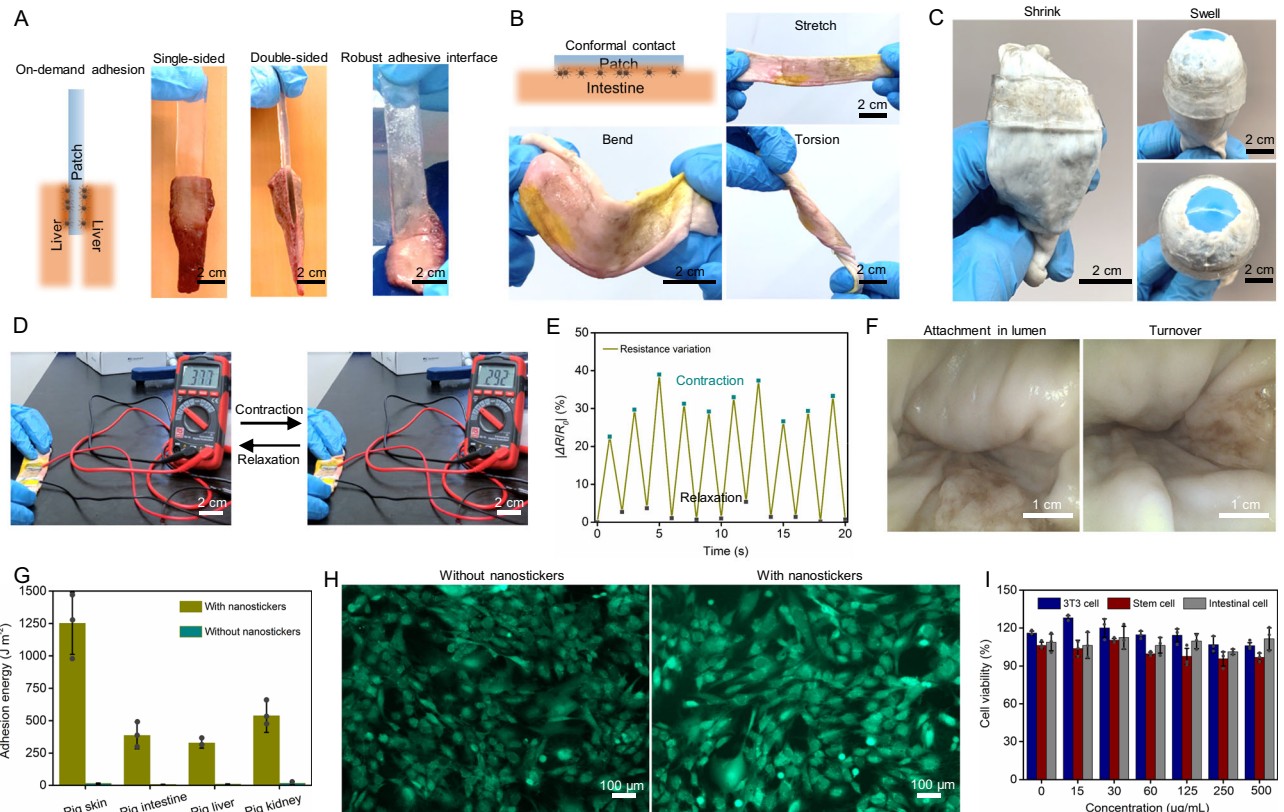

**Fig. 4 | Application of nanostickers on various tissues for magnetic control bioadhesion. A** Bioadhesive properties can be controlled over a wide range. **B** Stable and conformal adhesion enables the nanostickers-anchored patch to tolerate various mechanical deformation. **C** An intestinal lumen inserted with an inflatable ball is used to explore the extensibility and compliance of the anchored patch. **D** Resistance responses of the patch during the contraction-relaxation movements. **E** Resistance variations during repeated contraction-relaxation cycles. **F** Endoluminal delivery of the nanostickers for patch attachment. (**G**) Optimized adhesion energy by magnetic control bioadhesion on diverse tissues. Data are shown as mean ± SD; $n = 3$ independent samples. **H** Live/dead staining assay after 24 h of culture in NIH 3T3 fibroblasts shows that cells cultured with nanostickers have similar morphology and viability to cells cultured without nanostickers. **I** In vitro biocompatibility of the nanostickers assessed by cell proliferation assay using mesenchymal stem cells, NIH 3T3 fibroblasts, and intestinal epithelial cells after 72 h of culture. Data are shown as mean ± SD; $n = 3$ independent samples.

the highest adhesion energy is achieved with 10 min of steering and 3 Hz of rotational frequency, also displaying that the adhesive properties can be temporally controlled by the duration of magnetic treatment and the frequency of rotating magnetic field (Supplementary Fig. 15). Rotating nanostickers in a system with a low Reynolds number such as our case can generate a strong flow disturbance due to the local convection[39,40], wherein the magnetic and hydrodynamic drag forces and their torques synergistically control the motion of nanostickers (Supplementary Fig. 16). We observe that the nanostickers are dispersed randomly on the skin when there is no magnetic field, while the increased rotating frequency of the magnetic field leads to different assembled morphologies (Supplementary Fig. 17). This phenomenon is similar to the formation of micro/nanorobotic swarms in many dynamic fluids[41]. Therefore, we hypothesize that the gradient force pulls the nanostickers toward the magnet and the torque-induced vortex assists in assembling the nanostickers on the skin. To prove our hypothesis, we prepare nanostickers with high-viscosity chitosan (200-600 mpa.s) to create strong hindrance for their motion and assembly. The high-viscosity nanostickers provide much lower adhesion energy than low-viscosity nanostickers, suggesting that the gradient force and torque take important roles in anchoring of tissues (Supplementary Fig. 18). The motion and shear stress around the nanostickers are next simulated to help illustrate the anchoring mechanism. The nanostickers present negligible velocity and shear stress in water without magnetic control (Fig. 3E and Supplementary Fig. 19A). However, velocity and shear stress increase dramatically around the nanostickers when a rotating magnetic field is applied.

Meanwhile, the induced flow rate and shear stress are both improved by the increased length of assembled nanostickers (Fig. 3F and Supplementary Fig. 19B). According to these results, we can confirm that the magnetic field can actively steer the nanostickers for propulsion at the Z-axis direction and assembly on the XY plane, enabling to precisely control the nanostickers for tunable bonding on tissue surfaces (Supplementary Fig. 20).

## Selective bioadhesion on various tissues

Different viscous biofluids are used to contaminate pig skin to examine the propulsion of nanostickers in complex environments (Supplementary Fig. 21). The magnetic nanostickers can be effectively propelled through external barriers for robust bioadhesion (Supplementary Fig. 22). It is noteworthy that the nanostickers enable the patch to have on-demand adhesive properties. As a demonstration, the single-sided patch generally shows opposite adhesive properties while the double-sided patch can adhere to two pieces of pig livers simultaneously (Fig. 4A). For applications in biological tissues, the mechanical mismatch between the patch and tissues can accelerate adhesive failures and increase the risk of inflammation[42]. However, we can actively control the adhesive properties of nanostickers on different tissues and the thickness of the patch for conformal adhesion. For example, nanostickers with an area density around 0.5 µg/mm² make a thin patch (thickness ~100 µm) highly compliant with the pig intestine, showing excellent tolerance to external forces and torques (Fig. 4B). Surround patching also demonstrates that the adhesion can withstand the contraction and expansion of the intestine, as occurs in

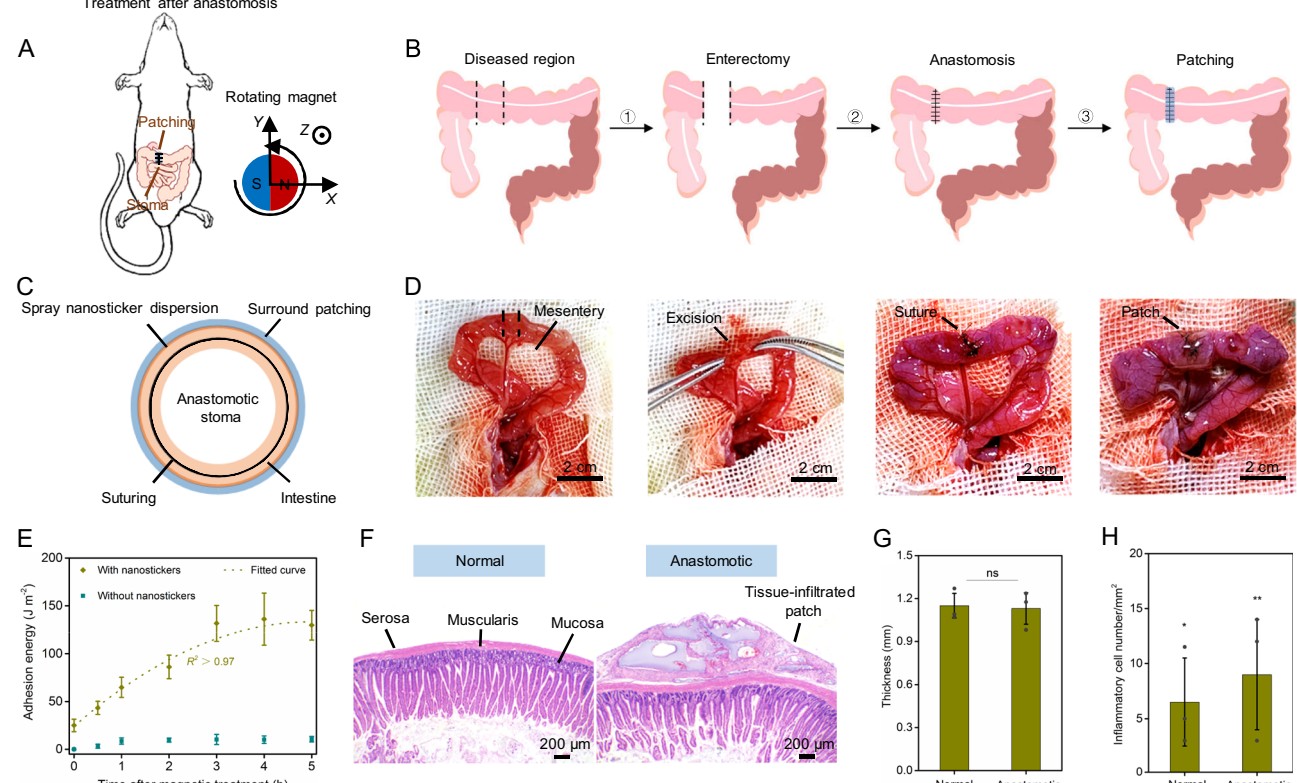

**Fig. 5 | Magnetic control bioadhesion assists disease treatment. A** Schematic illustration of the magnetic control of nanostickers in anchoring the anastomotic stoma for patching. **B** Entire treatment procedures for the diseased intestine (1: Excision of the diseased part; 2: Anastomose of the stoma by sutures; 3: Anchoring the stoma with magnetic nanostickers). **C** Sectional view of the nanostickers-anchored intestine and circular patch. **D** Images of each treatment procedure. **E** Adhesion energy varies over time after magnetic treatment. The fitted curve denotes the variation tendency. All values are presented as mean ± SD for $n = 3$

independent experiments. **F** Representative hemotoxylin and eosin staining images of the normal and patched groups. **G** Thickness of the intestinal wall in the normal and patched groups. Data are presented as mean ± SD; $n = 3$ independent samples. Statistical analyses are performed by using multiple two-sided $t$ tests with Bonferroni correction, *ns* represents not significant. **H** Histopathologic analysis of the number of inflammatory cells in the normal and patched groups. Data are presented as mean ± SD; $n = 3$ independent samples. Statistical analyses are performed by using two-sided $t$ test, *$P = 0.0196$, **$P = 0.0013$.

starvation and fullness (Fig. 4C). The conformal contact with the pig intestine suggests that the patch could serve as a good bioelectronic sensor to reflect the intestinal condition. During multiple cycles of contraction and relaxation, the patch shows rapid electrical responses ($\Delta R/R_O$) to monitor the small movement of the intestine (Fig. 4D and Supplementary Movie 3). The real-time resistance variations indicate that the contact between the intestine and patch is highly stable (Fig. 4E). As the nanostickers are steered remotely, we also demonstrate that magnetic control bioadhesion can be achieved in the intestinal lumen (Fig. 4F). The nanosticker dispersion is delivered into the lumen by a catheter and then actuated by magnetic field for anchoring. Following that, the thin patch is attached on the target region with the help of an endoscope (Supplementary Fig. 23A). The patch tightly adheres to the internal surface even under continuous turning, displaying a very stable adhesive interface (Supplementary Fig. 23B). Quantitative measurement confirms that the nanostickers can promote bioadhesion between the patch and different wet tissues by remote control, while the patch presents very low adhesion when applied without nanostickers (Fig. 4G). As the main components of the prepared nanostickers are approved by the Food and Drug Administration (FDA)[43,44], the nanostickers show excellent biocompatibility with various mammalian cells (Fig. 4H, I).

## Magnetic control bioadhesion in surgical operations
The remotely controlled properties enable the magnetic nanostickers to have significant potential in bridging fragile parts (e.g., diseased regions) or deep tissues with functional patches in vivo. Enterectomy

and anastomosis are frequently used to treat many diseases such as tissue necrosis, tumors, ischemia, and trauma[45–47]. For postoperative management, the tissue patch can prevent the migration of biofluids from the surgical site, as leakage can cause serious bacterial infections. Additionally, intense inflammation in the diseased region can induce complications such as tissue adhesion. The covered patch can therefore provide adhesion prevention between the diseased parts and other tissues. We developed an excised intestinal tract model in Sprague-Dawley rats to implement the magnetic control method and examine its feasibility and therapeutic effect (Fig. 5A). The excellent adhesive properties and biocompatibility demonstrate that magnetic control bioadhesion is applicable to postoperative treatment, which requires long-time functionality (Supplementary Fig. 24). The nanostickers are spread and steered after the suture-based anastomosis, providing controlled anchoring for the thin patch (Fig. 5B). Despite the sutured anastomotic stoma, small gaps between the two ends of the intestines may still exist. Spraying an appropriate concentration of nanosticker dispersion (~500 µg in total) around the stoma helps the patch fully cover these gaps with controlled and suitable adhesive properties, also avoiding harmful intestinal stenosis due to overly strong adhesion (Fig. 5C). To further mitigate the risks of infection and inflammation, a levofloxacin-loaded PAAm-Alg patch (thickness ~100 µm) is applied on the sutured stoma, showing stable and compliant bioadhesion (Fig. 5D and Supplementary Fig. 25). After attaching the patch, the adhesion energy increases and reaches a steady state around 3 h, allowing for repositioning the patch if it is misplaced initially (Fig. 5E). The patched groups show a survival rate at 100%, in

striking contrast to the high mortality rate of the sutured groups without patch treatment (Supplementary Fig. 26). Histomorphological analysis is performed to inspect the healing of the anastomotic stoma after 10 days of treatment. The patched groups display comparable intestinal structures to the normal groups (Fig. 5F), with a similar thickness of the intestinal wall (Fig. 5G). The inflammatory response in the patched groups is also close to the normal results (Fig. 5H). Furthermore, the intact intestine-patch interface for many days facilitates cell migration and tissue integration, evidenced by the strong infiltration of blood vessels and collagen fibers in the patch (Supplementary Fig. 27).

## Discussion

In summary, we have developed a strategy to enhance the tissue-hydrogel interface by intelligently steering nanostickers on diverse tissues. The interfacial adhesive properties can be greatly boosted and precisely controlled by setting related magnetic parameters. The spatial and temporal control of anchoring by remote magnetic field has many advantages over other typical adhesion strategies, particularly for bioadhesion on fragile body parts and deep tissues. We verified the feasibility of magnetic control method using a 50-mm-diameter permanent magnet, and the method is universal to versatile nanostickers and hydrogel patches. However, it is possible that the controllability of nanostickers and their anchoring effects could be improved if the magnetic field was provided by a larger magnet. Compared to passive diffusion of bridging polymers into tissues for anchoring, the active control of untethered nanostickers for rotation and propulsion endows bioadhesion with intelligence and extends its applicable scenarios, serving as a strong alternate or adjunct for medical supplies and contributing to precise medical treatment. Nevertheless, further improvement on the uneven distribution of nanostickers caused by the intrinsic magnetic field gradient deserves further exploration, which could expand the controllability and adhesive properties.

## Methods

### Ethics statement

All animal experiments including the surgery operations and materials used were reviewed and approved by the Institutional Animal Care and Use Committee in Shenzhen TOP Biotechnology Co. Ltd. (approval number: TOP-IACUC-2024-0101), which were performed in compliance with the law on experimental animals by China Committee for Research and Animal Ethics.

### Chemicals and Materials

Ferric trichloride hexahydrate ($FeCl_3 \cdot 6H_2O$; 99%), Ferrous sulfate heptahydrate ($FeSO_4 \cdot 7H_2O$; ≥99%), Calcium chloride dehydrate ($CaCl_2 \cdot 6H_2O$; ≥99%), Hydrochloric acid (HCl; 37%), Ammonium hydroxide solution ($NH_3 \cdot H_2O$; 25%-28%), Chitosan (low viscosity: <200 mpa.s), gelatin ( ~ 250 g; Bloom), Oxalyl chloride ( ≥ 99%), Sodium tripolyphosphate (98%), Sodium alginate (viscosity: $200 \pm 20$ mpa.s), Agar (low gel strength; 700-900 g/cm$^2$), N,N′-Methylenebis(acrylamide) (BIS; 99%), N,N,N′,N′-tetramethylethylenediamine (TEMED; 99%), Ammonium persulfate (APS; 99.99%) were purchased from Aladdin Chemicals. Acrylamide (AAm, 99%), N-Isopropylacrylamide (NIPAM; ≥ 98%) were purchased from J&K Scientific Ltd. Polyethylenimine (PEI; $M_n \approx 10$ kDa, branched) was purchased from Sigma-Aldrich. Chitosan (viscosity: 200-600 mpa.s) was purchased from TCI (Shanghai) Development Co. Ltd. Adipose-derived mesenchymal stem cell, NIH 3T3 cell, and intestinal epithelial cell were provided by the Faculty of Medicine, The Chinese University of Hong Kong. Each cell line was identified by an experienced technician using related biological technology such as PCR assays. Pig tissues, including skin, intestine, liver, and kidney, were freshly obtained from the slaughterhouse. The bioabsorbable suture is purchased from Shanghai Pudong Jinhuan Medical Supplies Co. Ltd. Deionized (DI) water with a resistivity of 18.2 MΩ was used during the experiments. All commercially obtained chemicals and materials are used without further purification.

### Preparation of superparamagnetic particles

1.35 g of $FeCl_3 \cdot 6H_2O$ was dissolved with 0.83 mL of HCl in 5 mL of water. 0.69 g of $FeSO_4 \cdot 7H_2O$ was dissolved with 0.42 mL of HCl in 2.5 mL of water. The clear $Fe^{3+}$ and $Fe^{2+}$ solutions were then mixed under protection by argon. 4 mL of $NH_3 \cdot H_2O$ was injected into the mixed solution and stirred for 40 min. The black sediment was finally washed by DI water and ethanol after centrifugation at $11,180 \times g$ for 30 min.

### Preparation of Fe₃O₄@chitosan nanosticker dispersion

The preparation referred to the previous methods with some modifications[48]. 30 mg of superparamagnetic particles was slowly dropped into a 3 mL of chitosan solution, which was dissolved in HCl solution to get a pH of approximately 5.5. 1.5 mg of sodium tripolyphosphate dissolved in 1 mL of water was added dropwise into the mixture to assist with surface coating. The mixed solution was stirred at 1000 rpm for 30 min at room temperature. $Fe_3O_4$@chitosan nanostickers were obtained by washing three times with DI water and ethanol after centrifugation at $11,180 \times g$ for 30 min. The quantity of the nanostickers was weighed after drying in a vacuum oven. The $Fe_3O_4$@chitosan nanosticker dispersion was sonicated adequately (UC-650; 50% amplitude of power 650 W) and used at a constant concentration (1 wt%, 10 mg/mL) unless otherwise specified.

### Preparation of various cationic polymer-coated nanostickers

For preparing $Fe_3O_4$@PEI nanostickers, 25 mL of oxalyl chloride (10 mg/mL) was added to 10 mg of $Fe_3O_4$ dispersion with stirring for 24 h, and the precipitate was obtained by washing with DI water and centrifugation at $11,180 \times g$ for 30 min. After that, 5 mg of the precipitate was re-dispersed in 20 mL of water and mixed with 20 mL of PEI solution (10 mg/mL) for 24 h reaction. $Fe_3O_4$@PEI nanostickers were collected by washing with DI water and ethanol after centrifugation at $11,180 \times g$ for 30 min. Finally, they were sonicated for dispersion. For preparing $Fe_3O_4$@gelatin nanostickers, the methods referred to a previous report with some modifications[49]. 500 mg of gelatin was dissolved in 50 mL of DI water by vigorously stirring at 60 °C. After cooling down, the $Fe^{3+}$ and $Fe^{2+}$ solutions with molar ratio at 2:1 were added into the gelatin solution and kept stirring for 2 h. Following that, $NH_3 \cdot H_2O$ was added into the solution until the pH reached ~11. The black suspension was further stirred for 6 h and then washed with DI water and ethanol after centrifugation at $11,180 \times g$ for 30 min. A well dispersed $Fe_3O_4$@gelatin suspension was obtained by sonication.

### Preparation of different hydrogel patches

For preparing the PAAm-Alg patch, 1 g of AAm was dissolved in 5 mL of water and then stirred with 250 mg of sodium alginate for 12 h. The monomer solution was transparent and sequentially mixed with 70 μL of BIS (2%, wt%), 30 μL of TMEMD, and 760 μL of APS (70 mg/mL) for mixing. The precursor solution was uniformly dropped onto a 10 cm × 10 cm glass mold and rapidly immersed into a $CaCl_2$ solution (20%, wt%) for 12 h. The thickness of the patch could be controlled by the applied amount of precursors. For preparing the PAAm-Agar patch, 250 mg of agar powder was dissolved in water by stirring at 90 °C, followed by blending with 1 g of AAm solution with 30 μL of BIS (2%, wt%), 10 μL of TMEMD, and 300 μL of APS (70 mg/mL). The mixture was rapidly dropped onto a glass mold with solidification for 12 h. For preparing the PNIPAM-Alg patch, 1 g of NIPAM and 250 mg of sodium alginate were dissolved in water and stirred for 12 h. Afterwards, they were added with 50 μL of BIS (2%, wt%), 20 μL of TMEMD, and 500 μL of APS (70 mg/mL). The mixture was dropped onto a glass mold and rapidly immersed into a $CaCl_2$ solution (20%, wt%) for 12 h at an ice bath.

## Transmission electron microscope (TEM) imaging of Fe₃O₄@chitosan nanostickers

The adequately sonicated nanosticker dispersion was dropped onto copper grids. After drying in the ambient, the copper grids were directly observed by transmission electron microscope (FEI/Philips CM-20) to adjust the magnification.

## Magnetic control bioadhesion

Tissues obtained from freshly slaughtered swine were used to implement the magnetic control strategy. Different quantities of nanosticker dispersions were uniformly spread (sprayed, dropped, or brushed) onto the designated region. Then, a 50-mm-diameter sphere permanent magnet was placed under the tissues at a certain distance. The rotating frequency and duration of magnetic treatment were set to control the motion and anchoring of nanostickers on tissue surfaces. Unless otherwise specified, the distance between the top surface of the magnet and the tissue surface was 3 cm, the rotating frequency was 3 Hz, and the duration of magnetic treatment was 10 min.

## SEM imaging of nanostickers on pig skin

After magnetic control of nanostickers for anchoring on pig skin, the pig skin was lyophilized. The freeze-dried pig skin was directly observed (accelerating voltage at 3 kV) by scanning electron microscope (S-4800, Hitachi) to adjust the magnification.

## AFM investigations on adhesion force

Tissues with/without anchored nanostickers were directly measured by atomic force microscope (Dimension ICON, Bruker). The AFM probe (RTESPA-150) with a tapping mode was applied that could continuously record the force and distance for its extension and retraction in the surface. All data were measured with a scan area at least 500 nm × 500 nm.

## Adhesion energy tests ex vivo

180° peel tests were performed using INSTRON-5566 according to the ASTM D2256 standard. A rigid polyethylene terephthalate film was adhered to the backside of the patch with cyanoacrylate glue to prevent large extension during peeling. The patch was attached on different tissues after controlling the nanostickers to complete the anchorage. Peel tests were conducted after 3 h of attachment for most evaluations unless otherwise stated. The patch attached on tissues without nanostickers served as the control group. One mechanical clamp was fixed to the end of tissues and the other mechanical clamp was fixed to the end of patch. The loading speed was constant at 1 mm/s. Real-time force-displacement data were recorded by the load cell and the adhesion energy was evaluated by two times of the steady force ($F$) divided by the width of patch ($W$). Each evaluation was repeated three times to acquire the average value of adhesion energy.

## Shear strength tests ex vivo

A patch was attached on the tissues with or without the anchorage of nanostickers. The anchored area was 2 cm × 2.5 cm (width × length) to ensure adequate connections between the patch and the tissues. Lap-shear measurements were then performed with a constant speed of 1 mm/s to evaluate the uniaxial tensile force-displacement data using the INSTRON-5566 machine. Shear strength was evaluated according to the tensile force divided by the adhesive area.

## Interfacial fatigue resistance tests

Cyclic 180° peel tests were performed by setting the cycle number ($N$) and applied force ($F_c$) in the INSTRON-5566 machine. The applied peel force was less than the steady-state peel force, and the interfacial crack extension ($c$) was recorded with cycle number $N$ to calculate the $d_c/d_N$. Different energy release rates $G$, defined as $F_c/W$ ($W$ is the width of the patch), were applied to plot the $d_c/d_N$ along with $G$. The interfacial

fatigue threshold $\tau_O$ was calculated by linearly extending the plot to the intercept of the $G$ axis.

## Simulations of magnetic field and motion of nanostickers

Simulations were conducted with COMSOL Multiphysics 5.5. For the magnetic field simulation, the magnet was modeled as N52-grade NdFeB with a spherical shape and a diameter of 50 mm ($d_{magnet}$ = 50 mm), with the surrounding environment modeled as air. For the simulation of nanostickers' motion, which would induce the flow field and pressure by the magnetic steering, the liquid medium was assumed to be water, and laminar flow was applied. The walls were defined as non-slipping boundaries. To illuminate the variation trend of different assembled morphologies, the chain-like magnetic nanostickers were modeled as rigid balls with a diameter of 100 nm. A rotation domain with a dynamic mesh was used to simulate the rotational motion of magnetic nanostickers with different chain lengths. The rotation frequency was set to 3 Hz and the snapshots were extracted at $t$ = 1 s.

## Electrical response investigations

Nanostickers with an area density around 0.5 μg/mm² were steered for anchoring on the pig intestine, and then a thin patch with a thickness ~100 μm was attached. Tinfoil was used to help immobilize the wire on both ends of the patch, which was also sealed with polyimide tapes. The wires were gripped by the probes of ohmmeter to record the real-time resistance variations when the intestine was moved.

## Endoluminal delivery of nanostickers for bioadhesion

The nanosticker dispersion was delivered into the lumen of the pig intestine by a catheter under the guidance of an endoscope. Then, the sphere permanent magnet was placed 3 cm under the intestine and actuated at a rotating frequency of 3 Hz for 10 min. Afterwards, a thin patch was delivered and applied through the endoscope. The attached patch on the intestine was rotated by continuous movement to examine the stability of adhesive interface. Snapshots was captured by the camera of endoscope.

## Live/dead staining assay

Calcein-AM and Propidium Iodide kits were used to determine the cellular state. 10⁴ of cells in a confocal dish were incubated with nanostickers to get a concentration at 500 μg/mL for 24 h. After that, the cell medium was discarded and the cells were washed by fresh medium to remove the extra nanostickers. Calcein-AM and Propidium Iodide (2 μL) were added into the cells and incubated at 37 °C for 30 min. Live/dead staining was then observed by fluorescence microscope (model).

## Biocompatibility evaluations

Mesenchymal stem cells, NIH 3T3 cells, and intestinal epithelial cells were used to evaluate the cell viability. 10⁴ of cells with 100 μL of medium in each well were cultured in different 96-well microplates. Nanostickers were added into the culture medium and subsequently transferred into the cell medium to get different concentrations from 0 μg/mL to 500 μg/mL for 72 h of incubation. The above medium was then discarded, and each well was added with 100 μL of MTS solution (1 mg/mL in fresh medium). After 2 h, the supernatant was removed and replaced with 100 μL of dimethyl sulfoxide (DMSO). After shaking the microplates for 1 min, cell viabilities could be evaluated by comparing the absorbance at 490 nm. In addition, cell viabilities without nanostickers were set as control groups.

## Drug release investigations

To demonstrate the localization of the drug and nanostickers, rhodamine 6 G was modeled as the drug loaded in the patch, and nanostickers were labeled with fluorescein isothiocyanate (FITC). The FITC-labeled nanostickers were prepared by reacting 1 mg of nanostickers

with 2 mmol of FITC for 24 h, followed by washing with DI water after centrifugation at $11,180 \times g$ for 30 min. The rhodamine-loaded patch was prepared by mixing the precursor solution of the patch with 2 mmol of rhodamine 6 G and then solidifying. After the FITC-labeled nanostickers were actuated to anchor on the tissues, the rhodamine-loaded patch was attached. The release tests were observed using a confocal fluorescence microscope (model).

## Magnetic control bioadhesion for treating fragile tissues

Intestinal tract injury models were implemented in female Sprague-Dawley rats ( ~ 300 g and ~12 weeks old) purchased from Zhuhai BesTest Bio-Tech Co. Ltd., and they were used to investigate the applicability of the magnetic control strategy. All rats were housed in separate cages and fed with sufficient food (e.g., glucose solution), and the environment was well ventilated with temperature and relative humidity were 25 °C and 60%, respectively. All surgical equipment was sterilized by iodophor solution. The rats were anesthetized by isoflurane, followed by shaving the hair on the abdomen under the help of 7% sodium sulfide solution. Afterwards, the abdomen was disinfected with iodophor and opened to expose the intestines. The intestines were excised with a length of 1 cm and a width of 0.5 cm by enterectomy to create the injury until approaching the mesentery. Following that, the two ends of the cut intestines were carefully aligned and anastomosed using bioabsorbable sutures to prevent stenosis or leak. 500 uL of nanosticker dispersion (1 wt%) was uniformly sprayed onto the anastomotic stomas and then actuated by a 50-mm-diameter sphere permanent magnet below the stomas. After successful anchorage, a levofloxacin-loaded ( ~ 200 mg) thin patch with a thickness of ~100 μm was attached to surround the stomas. Finally, the patched intestines were replaced into the abdomen, followed by suturing the separated tissues. The suture-only groups without patch treatment were set as comparison groups, for which the surgery operations on the excised intestinal tract models were the same. After carefully aligning and anastomosing the ends of the cut intestines by the bioabsorbable sutures, iodophor solution was used to disinfect and decrease inflammation of the stomas. The anastomotic intestines were then put back into the abdomen, followed by suturing and disinfection of the separated tissues. In daily care, the rats were fed with glucose solution (5 wt%) to help recovery, and the health status such as the behaviors and body temperature was regularly monitored. Humane endpoints would be offered according to the health status of rats, mainly including quick weight loss of 25%, decreased body temperature over 4 °C, continuous and deep malaise, convulsion, paralysis, and difficult breathing. 3 R principles were strictly obeyed during the experimental period.

## Histological assessment

All rats were sacrificed, and intestinal histology specimens were obtained on day 10. Healthy intestines were set as the normal groups. The vertically sectioned intestines under the patch were set as treated groups, ensuring the acquisition of the anastomotic region. The samples were preserved in 4% paraformaldehyde solution for fixation. To perform the histomorphological analysis, the samples were embedded in paraffin and cut within 5 μm in thickness and then they were stained by hematoxylin-eosin. The stained structures were observed by an optical microscope. The thickness of the intestinal wall includes the serosa, muscularis, and mucosa, and the number of infiltrated inflammatory cells in the normal and patched groups were compared to determine the healing efficacy.

## Statistical analysis

All results were shown as mean ± standard derivation via at least three samples unless otherwise noted. Statistical significance tests were performed using t test function in Origin 9.0 software. $P$ values less than 0.05 were regarded as statistically significant among the compared samples.

## Adhesion energy tests in vivo

Rats were performed with the same surgery operations and treatment. The sutured intestines without anchoring of nanostickers and attached by the patch were set as normal groups. The rats were sacrificed at different time to take out the intestine-patch hybrids from the abdomen, including the nanostickers-anchored patches and the normal patches. 180° peel tests were then performed to investigate the adhesion energy variations.

## Reporting summary

Further information on research design is available in the Nature Portfolio Reporting Summary linked to this article.

# Data availability

The data that can support the finding of this research are available in the main text and Supplementary Information. All data underlying this study are also available from the corresponding author upon request. Source data are provided with this paper.

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

## Acknowledgements

The authors acknowledge the support by the Innovation and Technology Fund (No. PsH/040/23) (C.H.), Hong Kong Research Grants Council (RGC) with project Nos. RFS2122-4S03, R4015-21, N_CUHK472/24, GRF 14209024, GRF 14205823, GRF 14301122, GRF 14300621, GRF 14203123, Strategic Topics Grant (project No. STG1/E-401/23-N), and the CUHK internal grants (L.Z.). The authors also thank the support from Multi-Scale Medical Robotics Centre (MRC), InnoHK, at the Hong Kong Science Park, the SIAT-CUHK Joint Laboratory of Robotics and Intelligent Systems and Li Ka Shing Institute of Health Sciences. The authors finally thank Dr. Wenqing He and Dr. Liqing Ai to help on materials characterization and biocompatibility evaluations.

## Author contributions

C.H. conceived the project, carried out the experiments, analyzed the data, and wrote the manuscript. J.G. assisted the surgical operation in rats. B.S. performed the dynamic simulation of nanostickers. K.F.C. and X.S. contributed to the supervision of experiment and editing the manuscript. L.Z. reviewed and edited the manuscript. All authors contributed to data interpretation and discussion in the manuscript.

## Competing interests

The authors declare no competing interests.
