## [Transparent Peer Review file · Nature Communications]

Magnetic nanostickers for active control of interface-enhanced selective bioadhesion

Corresponding Author: Professor Li Zhang

Version 0:

Reviewer comments:

Reviewer #1

(Remarks to the Author)

This manuscript presents a novel approach to surface bonding between biological tissue and hydrogel through the use of magnetic nanostickers, demonstrating promising adhesion capabilities. The key findings suggest that magnetic nanoparticles coated with a polymer can be manipulated to achieve effective and reversible surface adhesion. The study includes detailed measure of the material's adhesion strength with different type of nanostickers and patch and also the impact of magnetic control on modulating adhesive properties. The study also highlights potential applications in the biomedical field, including bioadhesion on various tissues, as well as its contribution to enhancing disease treatment. This non-invasive method showed promise in bioadhesion and is of interest to readers of the Nature communications journal. Therefore, I would recommend that this study be published with minor revisions. Detailed minor comments follow.

The idea of using magnetic nanostructures to enhance adhesion is innovative. This strategy presents a remote control in adhesion energy and also give a spatiotemporal controllability, unlike conventional adhesion strategies (such as adhesive bonding and topological connections), which offer no control over space and time and exhibit poor interfacial fatigue resistance. In addition, their method can be applied to a wide range of body parts, including fragile or diseased regions, unlike ultrasound-mediated emerging methods, which, although they allow control over adhesion, are unsuitable for fragile body parts due to the use of thick ultrasound probes and high pressure.

I think a more detailed comparison is missing that highlights how their concept differs from the existing literature on nanoparticle-based adhesives.

Note that authors could comment on the following paper:

Meddahi-Pelle, A., A. Legrand, A. Marcellan, L. Louedec, D. Letourneur and L. Leibler (2014). "Organ Repair, Hemostasis, and In Vivo Bonding of Medical Devices by Aqueous Solutions of Nanoparticles." *Angewandte Chemie-International Edition* 53(25): 6369-6373.

The authors have conducted extensive experiments to validate their concept, including adhesion tests, shear strength tests, interfacial fatigue resistance tests on various tissue. The authors also conducted in vivo experiments to demonstrate the potential for real-world application of their concept in biomedical fields.

The methodology employed in this work is generally robust, and the data is presented clearly and as mean \pm standard deviation based on at least three samples.

Insufficient detail on nanostickers synthesis and characterization: Synthesis following a protocol from a publication; if so, should it be mentioned if no a specific characterization of nanoparticles is important? More specific data regarding their size, surface morphology and chemistry, composition, dispersion of particle? Particle coated polymer like thickness of coating? Spreading and anchoring mechanism of nanostickers not clear?

These specifications would help readers understand the system more accurately. This will further lead to a better understanding of this study.

Adhesion mechanism: The manuscript mentions that the stable tissue-patch interface is achieved through multiple interactions from the nanostickers, including strong magnetic attraction and crosslinks with the robust patch but the underlying mechanism is not fully explained. The authors should include a more thorough discussion of the type of interaction existing between the different component of the material.

They synthesize different types of nanostickers, but they do not explain the mechanisms underlying the differences in adhesion between these particles.

Comment on the differences in adhesion energies of the various hydrogels: Are these differences due to specific interactions, or are they simply a result of the mechanical properties of the different patches?

Specific Comments:

- Specify rate of loading for shear strength tests are required.
- Write the full names of acronyms when they showed up for the first time (e.g., NIPAM, FITC, DI...).
- The authors reference methods such as SEM, TEM, and AFM, but these techniques and the parameters used, are not included in the experimental section.
- Specification of type of nanostickers, area density, patch type and thickness in some studies.
- The quantity of the nanostickers was weighed after drying in a vacuum oven.

They demonstrate that magnetic control bioadhesion can help in surgical operations by conducting in vivo experiments on post-treatment of diseased intestinal regions after enterectomy and anastomosis.

Reviewer #2

(Remarks to the Author)

The authors successfully demonstrate a method of attaching patches to tissues via controlled embedding of magnetic nanostickers into the tissue. The nanostickers increase the interfacial bonding, which the authors also investigated via several methods.

The paper may be interesting to the readership of Nature Communications, however several issues have to be addressed before the publication can be recommended. The main objection is the absence of a clear flow of the argument. There seems to be many details which distract the reader from the main idea and the understanding of the underlying mechanism. A major revision is recommended.

Please find more detailed comments below:

- 1) Abstract and many other places in the main text: Is the word "spaciotemporal" required? What is the meaning of this expression? Please explain in the text.
- 2) L. 18: "adhesion-related space and time". Please explain.
- 3) L. 49: Please quantify the minimal bonding energy.
- 4) L. 77, 142: Which magnetic field is gradient based? Do you mean non-uniform magnetic field? Please quantify the gradients of magnetic fields.
- 5) L. 158 and below: The description of the anchoring mechanism seems to be not very persuasive. It looks like a speculation because it is not supported by any theory. Could you please formulate a clearer physical picture of observed phenomena?
- 6) L. 159, 160: This sentence makes no sense. "Illumination of the motion"?
- 7) L. 166, Fig. 3B: "displays that the magnetic field distribution is uniform on the XY plane". Fig 3B clearly shows that the field is non-uniform. Please specify or correct. Furthermore, the smallest distance on Fig. 3B is 35 mm, but the method needs 30 mm. Why is there a discrepancy?
- 8) L. 185: Is it the magnetic force or the torque or maybe both? Could you please quantify the torque?
- 9) L. 257: The authors describe a use case for deep tissues, while the method requires the magnet to be placed 3 cm below the tissue (L. 335). Does this mean that the magnet should be positioned inside the body? Is this realistic? The authors provide a valid use case for surgery in the intestines, however there are limitations. Please comment on this.
- 10) Fig. S7 is in EMU and CGS units, while S2 and S3 is in SI units. Please bring all figures to one unit system.
- 11) Figure S7: Is it external or internal magnetic field? If it is internal field, how is it calculated? If it is external field, what is the size of the sample and its composition? Also, Fig. S7 might refer to the mass magnetization.
- 12) Figure S6: Information on the vertical axis is missing.
- 13) Figure S15: Why is the magnetic scalar potential shown? It is not measurable. It is trivial that the magnetic field from a spherical permanent magnet is non uniform. The purpose of this figure is not clear.
- 14) Supplementary L. 136: The formula for the magnetic force on the magnetic dipole seems to be not correct. It seems to be a formula for the electric dipole. It should be $F = \nabla(m \cdot B)$.

Reviewer #3

(Remarks to the Author)

Reviewer #4

(Remarks to the Author)

The manuscript proposed an interesting alternative approach to manipulate bioadhesion using nanostickers and magnetic

fields. The authors systemically analyzed the effects of magnetic fields on the nanostickers and their interactions with biological tissues. They further demonstrated the advantages of the nanostickers for facilitating anastomosis and recovery in vivo.

The manuscript was well written. I would recommend consideration of the manuscript, after addressing the following comments and questions.

(1) The authors claimed that they can enable the “spatiotemporal” anchoring of the nanostickers, which is one of the potential advantages of the proposed remote magnetic control compared to other methods, but the evidence presented in the current manuscript is not convincing. It is clear that the adhesion energy can be mediated by the applied magnetic field, however, the authors need to provide more evidence to highlight the proposed approach for controlling adhesion in space and in time.

- The spatial control was achieved mainly through “masking”, which seems to be physically blocking the contact between the nanosticker and the tissue, but not via controlling the magnetic field. More evidences need to be provided to show that the pattern can be directly regulated by magnetic forces, particularly considering the uneven distribution and clustering of the nanoparticles acknowledged by the authors.

- No data is provided to demonstrate the so-called temporal control. The authors need to show data illustrating the adhesion, and importantly the debonding, can be modulated by the magnetic field, if possible.

Otherwise, although the authors have highlighted the significance of the spatiotemporal control of the bioadhesion throughout the manuscript, the results in the current manuscript do not support the claimed “spatiotemporal control of anchoring by remote magnetic field”.

(2) The authors demonstrated the different “adhesion force” of the nanostickers using AFM. Can the authors provide more details on the AFM probe used? What is the surface mechanical and chemical properties of the probe? How is this set of measurement relevant to the bioadhesion between the hydrogel, the nanoparticle, and the tissue?

(3) The authors demonstrated the application of the nanostickers in GI tract, which often involve various pH levels. How will the bioadhesion energy of the nanostickers be affected by the environmental pH? It has been shown that the pH can significantly affect the adhesion of chitosan-based bioadhesives.

(4) In the in vivo studies, the authors mentioned that a “levofloxacin-loaded (~200 mg) thin patch” was applied. What is the composition of such patch? Is it PAAm-alginate based hydrogel? What absorbable suture was used, since there’s no trace of suture remnant in the regions of interest shown in the histology images? What is the cause of death for animals that did not survive the surgeries (suture-only groups)?

(5) The authors claimed that the “long-term integration with fragile tissues” was demonstrated, but only 10-day short-term implantation was conducted. Unless the author can demonstrate the biointegration of tissue-nanostickers for months, the reviewer suggests the authors to adjust their claims.

Version 1:

Reviewer comments:

Reviewer #1

(Remarks to the Author)

The authors have provided satisfactory point-by-point answers. I therefore recommend publication of the manuscript as is.

Reviewer #2

(Remarks to the Author)

The authors have replied to my comments properly. I have only two remarks. The authors should check the consistency of the usage of the physical quantities related to magnetic field: the magnetic field strength H which unit is A/m and the magnetic flux density B which unit is T. The authors are recommended to check the consistency of the usage of the ∇ symbol for the magnetic force.

Reviewer #3

(Remarks to the Author)

Reviewer #4

(Remarks to the Author)

The reviewer’s questions have been addressed. However, the methods section could be further improved by providing more details on the humane endpoints caused by the inflammation/complications.

Version 2:

Reviewer comments:

Reviewer #2

(Remarks to the Author)

It is still not clear how the authors work with the del operator, see the file attached.
The authors may wish to consult the paper of T.H. Boyer, Am. J. Phys. 56, 688–692 (1988).

Reviewer #3

(Remarks to the Author)

Reviewer #4

(Remarks to the Author)

The reviewer's comments have been addressed. The acceptance of the manuscript is suggested.

Version 3:

Reviewer comments:

Reviewer #2

(Remarks to the Author)

I want to refrain from making a conclusion about the quality of the revision.

Reviewer #3

(Remarks to the Author)

Reviewer #5

(Remarks to the Author)

[Note from the Editor: the report of Reviewer #5 is attached.]

Version 4:

Reviewer comments:

Reviewer #5

(Remarks to the Author)

[Note from the Editor: the report of Reviewer #5 is attached.]

Reviewer #1 (Remarks to the Author):

This manuscript presents a novel approach to surface bonding between biological tissue and hydrogel through the use of magnetic nanostickers, demonstrating promising adhesion capabilities. The key findings suggest that magnetic nanoparticles coated with a polymer can be manipulated to achieve effective and reversible surface adhesion. The study includes detailed measure of the material's adhesion strength with different type of nanostickers and patch and also the impact of magnetic control on modulating adhesive properties. The study also highlights potential applications in the biomedical field, including bioadhesion on various tissues, as well as its contribution to enhancing disease treatment. This non-invasive method showed promise in bioadhesion and is of interest to readers of the Nature communications journal. Therefore, I would recommend that this study be published with minor revisions. Detailed minor comments follow.

Response: We thank the reviewer's highly positive comments. We have responded to the comments point by point and revised the manuscript accordingly.

The idea of using magnetic nanostructures to enhance adhesion is innovative. This strategy presents a remote control in adhesion energy and also give a spatiotemporal controllability, unlike conventional adhesion strategies (such as adhesive bonding and topological connections), which offer no control over space and time and exhibit poor interfacial fatigue resistance. In addition, their method can be applied to a wide range of body parts, including fragile or diseased regions, unlike ultrasound-mediated emerging methods, which, although they allow control over adhesion, are unsuitable for fragile body parts due to the use of thick ultrasound probes and high pressure.

Response: Thanks a lot for the high praise of our work such as "The idea of using magnetic nanostructures to enhance adhesion is innovative" and "their method can be applied to a wide range of body parts, including fragile or diseased regions, unlike ultrasound-mediated emerging methods, which, although they allow control over adhesion, are unsuitable for fragile body parts due to the use of thick ultrasound probes and high pressure".

I think a more detailed comparison is missing that highlights how their concept differs from the existing literature on nanoparticle-based adhesives.

Note that authors could comment on the following paper:

Meddahi-Pelle, A., A. Legrand, A. Marcellan, L. Louedec, D. Letourneur and L. Leibler (2014). "Organ Repair, Hemostasis, and In Vivo Bonding of Medical Devices by Aqueous Solutions of Nanoparticles." *Angewandte Chemie-International Edition* 53(25): 6369-6373.

Response: Thanks for the suggestion. We have provided the comparison on previous nanoparticle-based adhesives including the suggested paper to highlight the concept. The detailed comparison is provided on page 2 in the manuscript.

The authors have conducted extensive experiments to validate their concept, including

adhesion tests, shear strength tests, interfacial fatigue resistance tests on various tissue. The authors also conducted in vivo experiments to demonstrate the potential for real-world application of their concept in biomedical fields. The methodology employed in this work is generally robust, and the data is presented clearly and as mean \pm standard deviation based on at least three samples.

Response: Thanks for the positive comments on our experiments and methodology.

Insufficient detail on nanostickers synthesis and characterization: Synthesis following a protocol from a publication; if so, should it be mentioned if no a specific characterization of nanoparticles is important? More specific data regarding their size, surface morphology and chemistry, composition, dispersion of particle? Particle coated polymer like thickness of coating? Spreading and anchoring mechanism of nanostickers not clear? These specifications would help readers understand the system more accurately. This will further lead to a better understanding of this study.

Response: Thanks for the comment. In this work, the synthesis of nanostickers is not the main advantage, for which we referred to previous methods with some modifications. For better presentation, we have added the references related to the synthesis in the manuscript. This study focuses on investigating the magnetic control method for controlled bioadhesion, and it is applicable to different types of nanostickers (Supplementary Fig. 2) and hydrogel patches (Supplementary Fig. 3). Therefore, the main influence on the controlled anchoring of nanostickers should not be attributed to the advantages of nanoparticle synthesis. In addition, we have performed many characterizations to investigate the size, morphology, chemical components, surface potential, and magnetization (Supplementary Figs. 4-8) for representative Fe_3O_4 @chitosan nanostickers, which may be sufficient to support the claims in the manuscript. For the spreading and anchoring mechanism of nanostickers, we have also supplemented a scheme (Supplementary Fig. 20) and descriptions for better understanding of this work.

Adhesion mechanism: The manuscript mentions that the stable tissue-patch interface is achieved through multiple interactions from the nanostickers, including strong magnetic attraction and crosslinks with the robust patch but the underlying mechanism is not fully explained. The authors should include a more thorough discussion of the type of interaction existing between the different component of the material.

Response: Thanks for the suggestion. We have provided more discussions on page 5 to elucidate the interactions between different components.

They synthesize different types of nanostickers, but they do not explain the mechanisms underlying the differences in adhesion between these particles.

Response: Thanks for the comment. We have provided additional explanations on page 5 for the differences in adhesion between different nanostickers.

Comment on the differences in adhesion energies of the various hydrogels: Are these differences due to specific interactions, or are they simply a result of the mechanical

properties of the different patches?

Response: Thanks for the question. These hydrogel patches are prepared through similar polymerization methods, and the functional groups on biopolymers between various hydrogels are similar. Therefore, compared to the huge differences in the mechanical properties, it should be not significant for the variations in the interactions.

Specific Comments:

- Specify rate of loading for shear strength tests are required.

Response: We have specified the rate of loading for shear strength tests.

- Write the full names of acronyms when they showed up for the first time (e.g., NIPAM, FITC, DI...).

Response: We have written these acronyms into full names.

- The authors reference methods such as SEM, TEM, and AFM, but these techniques and the parameters used, are not included in the experimental section.

Response: According to your suggestion, we have supplemented these technique methods and related parameters in the experimental section.

- Specification of type of nanostickers, area density, patch type and thickness in some studies.

Response: We have specified the information in some studies.

- The quantity of the nanostickers was weighed after drying in a vacuum oven.

Response: Thanks for pointing this out. We have revised the typo accordingly.

They demonstrate that magnetic control bioadhesion can help in surgical operations by conducting in vivo experiments on post-treatment of diseased intestinal regions after enterectomy and anastomosis.

Response: Thanks for the detailed comments. We have revised the manuscript according to these comments.

Reviewer #2 (Remarks to the Author):

The authors successfully demonstrate a method of attaching patches to tissues via controlled embedding of magnetic nanostickers into the tissue. The nanostickers increase the interfacial bonding, which the authors also investigated via several methods. The paper may be interesting to the readership of Nature Communications, however several issues have to be addressed before the publication can be recommended. The main objection is the absence of a clear flow of the argument. There seems to be many details which distract the reader from the main idea and the understanding of the underlying mechanism. A major revision is recommended.

Response: We thank the reviewer's constructive comments. According to your suggestions, we have revised the manuscript and provided point-by-point responses to your comments.

Please find more detailed comments below:

1) Abstract and many other places in the main text: Is the word "spatiotemporal" required? What is the meaning of this expression? Please explain in the text.

Response: Thanks for the suggestion. The "spatiotemporal" aimed to indicate the ability of nanostickers in spatial and temporal control of anchoring. In other words, the nanostickers could be controlled to be anchored with different areas (Supplementary Fig. 9), and the time of magnetic treatment could control the anchoring effect of the nanostickers (Supplementary Fig. 15A and 15B). For clear description, we have revised the word and provided a related explanation in the text.

2) L. 18: "adhesion-related space and time". Please explain.

Response: The meaning of "adhesion-related space and time" was that the adhesion (anchoring) of nanostickers on tissues could be controlled by different areas and different time of magnetic treatment. For ease of understanding, we have revised the description in the text.

3) L. 49: Please quantify the minimal bonding energy.

Response: According to your suggestion, we have provided the value for reference. The interaction energy between a nanoparticle and different tissue membranes is reported to be around 5-10 hydrogen bonds, ranging from -4.4 to +5.1 kJ mol⁻¹ (*Environ. Sci.: Nano* **10**, 424-439 (2023)). The actual interaction energy may be varied with the physicochemical properties of materials and tissues.

4) L. 77, 142: Which magnetic field is gradient based? Do you mean non-uniform magnetic field? Please quantify the gradients of magnetic fields.

Response: Thanks for the question. The described gradient ($\Delta B/\Delta d_{mt}$) in the magnetic field is the variation of magnetic field strength (**B**) versus the vertical distance between the top surface of the magnet and tissue surface (d_{mt}). According to your suggestion, we have provided the magnetic field gradient at different d_{mt} (Fig. R1).

Fig. R1. The magnetic field gradient at different d_{mt} can be obtained by calculating the variation of the magnetic field strength versus d_{mt} .

5) L. 158 and below: The description of the anchoring mechanism seems to be not very persuasive. It looks like a speculation because it is not supported by any theory. Could you please formulate a clearer physical picture of observed phenomena?

Response: Thanks for the comment. We actually investigated the anchoring mechanism of nanostickers through both experimental and simulated methods. First, a rotating magnetic field with different parameters enabled the anchored nanostickers to have various adhesion energy (Supplementary Fig. 15), and the adhesion energy was extremely low at 26.8 J m^{-2} if no magnetic field was applied (Fig. 3D). These results substantiated that the magnetic fields could control the nanostickers for different adhesion energy. Following that, in order to investigate how the rotating magnetic field controls the nanostickers in three-dimensional XYZ directions, we calculated the magnetic field gradient and gradient-induced magnetic force at the Z -axis direction, as well as the magnetic torque of the nanostickers caused by rotation of the magnetic field on the XY plane (Supplementary Fig. 16), and the assembled morphology of the nanostickers in different rotating magnetic fields was observed (Supplementary Fig. 17). These results demonstrated that the remarkable magnetic field gradient at Z -axis direction could propel the nanostickers to tightly bond with the tissues while the magnetic field on the XY plane could control the aggregation of the nanostickers. Afterwards, we also created a high-viscosity environment, aiming to prevent the motion of the nanostickers for propulsion and aggregation, which caused much reduced adhesion energy for the anchored nanostickers (Supplementary Fig. 18). Here, the decreased adhesion energy verified that the anchoring effect of the nanostickers was highly related to the propulsion and aggregation of the nanostickers. We further simulated the nanostickers' motion steered by the magnetic field (Supplementary Fig. 19), because their motion could induce water flow and shear pressure (Fig. 3E and 3F). The simulated results demonstrated that there indeed exist high shear stress, particularly for the assembled nanostickers, for which the high shear stress could help the nanostickers to be anchored on the tissues. According to your suggestion, we have further provided a scheme for elucidating the anchoring mechanism (Fig. R2). We hope that these additional descriptions can help you consider the anchoring mechanism is supported by different experimental and simulated results.

Fig. R2. Schematic diagram for the anchoring procedure of nanostickers: 1. Spread nanosticker dispersion by different methods such as spraying, dropping, and brushing. 2. Actuate the rotating magnetic field to rotate nanostickers for assembly (XY plane) and propel nanostickers for bonding with tissues (Z -axis direction). 3. Nanostickers are successfully anchored when their motion is hampered by the rough tissues.

6) L. 159, 160: This sentence makes no sense. “Illumination of the motion”?

Response: Thanks for pointing this out. We have removed the sentence for better flow.

7) L. 166, Fig. 3B: “displays that the magnetic field distribution is uniform on the XY plane”. Fig 3B clearly shows that the field is non-uniform. Please specify or correct. Furthermore, the smallest distance on Fig. 3B is 35 mm, but the method needs 30 mm. Why is there a discrepancy?

Response: Thanks for the comment. As you indicated, the magnetic field from a spherical permanent magnet is non-uniform, while the change of field strength vertically above the magnet is very small (e.g., ~ 0.4 mT at $d_{mt} = 30$ mm) for X -coordinate from -25 mm to 25 mm (Fig. 3B), especially compared to the large changes at Z -coordinates (Supplementary Fig. 16B) and other X -coordinates. On the XY plane, the small change of field strength in this range guarantees that the nanostickers are primarily controlled by the magnetic torque for rotating as the magnetic field is rotational. There might exist an inaccurate description for the magnetic field distribution, according to your suggestion, we have revised it for a clear description. In addition, because the displayed distance in Fig. 3B is Z ($Z = d_{mt} + r_m$, where r_m is the radius of the magnet (25 mm)), the smallest distance between the top surface of the magnet and tissue surface (d_{mt}) at $Z = 35$ mm is 10 mm. The actual 30 mm stated in the method ($d_{mt} = 30$ mm) is equal to $Z = 55$ mm.

8) L. 185: Is it the magnetic force or the torque or maybe both? Could you please quantify the torque?

Response: Based on the experimental results, we consider that both magnetic force and magnetic torque can help the anchoring of nanostickers on tissues. As we described above, the remarkable magnetic field gradient at the Z -axis direction ensures that the nanostickers have sufficient magnetic force for propulsion (Supplementary Fig. 16), and the magnetic torque induced by the rotating magnetic field with different frequencies on the XY plane assists the assembly of nanostickers (Supplementary Fig. 17). Furthermore, high-viscosity nanostickers with weaker propulsion and rotation show much lower adhesion energy than low-viscosity nanostickers (Supplementary Fig. 18). The calculation of magnetic torque induced by the rotating magnetic field has been provided below:

The magnetic torque $\int_{V_{nanosticker}} \mathbf{M} \times \mathbf{B} dV_{nanosticker}$ can be simplified as $\mathbf{M} \times \mathbf{B}$ due to the small size ~ 10 nm of the nanosticker (*Adv. Intell. Syst.* **5**, 2200416 (2023)), in which \mathbf{M} is the total dipole moment of the nanosticker and \mathbf{B} is the flux density of the external field at the integral point. For a nanosticker standing vertically above the magnet at $d_{mt} = 30$ mm, $|\mathbf{B}|$ is equal to 0.045 (T) and $|\mathbf{M}|$ can be calculated to 35 ($\text{A}\cdot\text{m}^2\cdot\text{kg}^{-1}$) from the mass magnetization curve (Supplementary Fig. 7). Therefore, the magnetic torque can be formulated as $|\mathbf{T}_{m-nanosticker}| = 0.045 \times 35m_{nanosticker} = 1.58m_{nanosticker}$ ($\text{N}\cdot\text{m}$), where $m_{nanosticker}$ represents the mass of the nanosticker (kg). It is noteworthy that as the actual mass of each nanosticker differs due to their various sizes, the difference on the magnetic torque of each nanosticker may exist.

9) L. 257: The authors describe a use case for deep tissues, while the method requires the magnet to be placed 3 cm below the tissue (L. 335). Does this mean that the magnet should be positioned inside the body? Is this realistic? The authors provide a valid use case for surgery in the intestines, however there are limitations. Please comment on this.

Response: Thanks for the nice question. In the demonstration, we positioned the magnet beside the rats (Fig. 5A) and the intestines were taken out from the body (Fig. 5D). Because the intestines are long muscular tubes, the magnet and the intestines could be moved to match with the required distance. For other deep tissues, it may be difficult to take them out from the body, but a larger magnet (diameter > 50 mm) with stronger magnetization positioned below the rats can also provide the same magnetic parameters (e.g., field strength and field frequency) with the abovementioned magnetic field, and this is possible to directly steer the nanostickers for anchoring in body, as the anchoring of nanostickers is highly related to the imposed magnetic field.

10) Fig. S7 is in EMU and CGS units, while S2 and S3 is in SI units. Please bring all figures to one unit system.

Response: Thanks for pointing this out. We have revised the Supplementary Fig. 7 into SI units accordingly (Fig. R3).

Fig. R3. Magnetic hysteresis curves of superparamagnetic particles and Fe_3O_4 @chitosan.

11) Figure S7: Is it external or internal magnetic field? If it is internal field, how is it calculated? If it is external field, what is the size of the sample and its composition? Also, Fig. S7 might refer to the mass magnetization.

Response: The magnetic hysteresis curve was measured by the vibrating sample magnetometer (VSM). Taking advantage of the electromagnetic induction principle, VSM could implement a precise external magnetic field to vibrate samples, and then measure the magnetic torque of samples. The measured samples were dried Fe_3O_4 and Fe_3O_4 @chitosan nanoparticles, which were quantified with precise mass. As you said, the results in Supplementary Fig. 7 were mass magnetization curves.

12) Figure S6: Information on the vertical axis is missing.

Response: Thanks for pointing this out. We have added information on the vertical axis (Fig. R4).

Fig. R4. FTIR spectra of the main components in Fe_3O_4 @chitosan.

13) Figure S15: Why is the magnetic scalar potential shown? It is not measurable. It is trivial that the magnetic field from a spherical permanent magnet is non uniform. The purpose of this figure is not clear.

Response: Thanks for the comment. The magnetic scalar potential distribution was simulated through COMSOL Multiphysics 5.5 software. We showed the distribution for it might help readers to understand the applied magnetic field. In order to avoid misinterpreting the meaning, according to your suggestion, we have removed this figure.

14) Supplementary L. 136: The formula for the magnetic force on the magnetic dipole seems to be not correct. It seems to be a formula for the electric dipole. It should be $\mathbf{F} = \nabla(\mathbf{m} \cdot \mathbf{B})$.

Response: Thanks for the comment. The original formula of magnetic force on the magnetic dipole is indeed $\mathbf{F} = \nabla(\mathbf{M} \cdot \mathbf{B})$. Because the nanosticker (~ 10 nm) is quite small compared to the spatial variation of the external magnetic field, the nanosticker is considered as a dot and thus \mathbf{M} (the total dipole moment of the controlled nanosticker) can be regarded as a constant vector. The formula $\mathbf{F} = \nabla(\mathbf{M} \cdot \mathbf{B})$ can therefore be simplified as $\mathbf{M} \cdot \nabla \mathbf{B}$. More information about the formula of magnetic force can be found in our previous review (*Adv. Intell. Syst.* **5**, 2200416 (2023)). For better presentation, we have provided additional descriptions in the text for this deduction.

Reviewer #3 (Remarks to the Author):

Response: Thanks for the reviewer's comment. We have provided comprehensive responses to all the reviewers' comments point by point.

Reviewer #4 (Remarks to the Author):

The manuscript proposed an interesting alternative approach to manipulate bioadhesion using nanostickers and magnetic fields. The authors systemically analyzed the effects of magnetic fields on the nanostickers and their interactions with biological tissues. They further demonstrated the advantages of the nanostickers for facilitating anastomosis and recovery in vivo. The manuscript was well written. I would recommend consideration of the manuscript, after addressing the following comments and questions.

Response: We thank the reviewer's positive comments very much. Following please find our detailed responses to your comments point by point.

(1) The authors claimed that they can enable the "spatiotemporal" anchoring of the nanostickers, which is one of the potential advantages of the proposed remote magnetic control compared to other methods, but the evidence presented in the current manuscript is not convincing. It is clear that the adhesion energy can be mediated by the applied magnetic field, however, the authors need to provide more evidence to highlight the proposed approach for controlling adhesion in space and in time.

- The spatial control was achieved mainly through "masking", which seems to be physically blocking the contact between the nanosticker and the tissue, but not via controlling the magnetic field. More evidences need to be provided to show that the pattern can be directly regulated by magnetic forces, particularly considering the uneven distribution and clustering of the nanoparticles acknowledged by the authors.
- No data is provided to demonstrate the so-called temporal control. The authors need to show data illustrating the adhesion, and importantly the debonding, can be modulated by the magnetic field, if possible.

Otherwise, although the authors have highlighted the significance of the spatiotemporal control of the bioadhesion throughout the manuscript, the results in the current manuscript do not support the claimed "spatiotemporal control of anchoring by remote magnetic field".

Response: Thanks for the professional comments. We are sorry that there might cause a misunderstanding for the "spatiotemporal" anchoring of nanostickers. For the spatial control of anchoring, we considered that it represented the ability of controlling the nanostickers to be anchored in different areas. Indeed, if the pattern shape should be taken into account, the complex patterns could only be achieved with the help of masks, because the nanostickers could not be separated from the solvent (water) and thus the shape of anchored nanostickers was determined by the shape of water. However, using different spreading methods such as dropping, spraying, and brushing, we could make the nanosticker dispersion with simple shapes on tissues. For example, without using any masks, the dropped nanosticker dispersion could be directly steered to present simple circles (Fig. R5A), which were realized by dropping a droplet of nanosticker dispersion and then actuated by a rotating magnetic field. As we all know, water droplet is easy to form a circular contact surface despite various contact angles. In fact, we remotely controlled the nanostickers for anchoring on tissues without any masks in all

experiments except for the complex clover-shaped patterns. In order to describe accurately on the spatial control of anchoring, we have further revised the description in the text and provided a scheme for illustration (Fig. R5B).

Fig. R5. (A) Nanostickers can be anchored on pig skin within a circle by magnetic steering without using masks. (B) Schematic illustration of spatially controlling the nanostickers for simple patterns such as circle.

For the temporal control of anchoring, we aimed to demonstrate that the anchoring effect could be controlled by the time of magnetic treatment. According to your suggestions, we have provided the time-dependent work of retraction and adhesion energy (Fig. R6). These results indicate that the anchoring effect is increased with the time of magnetic treatment and reaches steady around 10 min. Yet, we have not found that the time of magnetic treatment could control the debonding of nanostickers, while we will further explore this point in the future study.

Fig. R6. (A) Time-dependent work of retraction through different time of magnetic steering of nanostickers on pig skin. (B) Time-dependent adhesion energy of attached hydrogel patch through different time of magnetic steering of nanostickers (area density: $2 \mu\text{g}/\text{mm}^2$) on pig skin.

We hope that these revisions can help to understand accurately the ability of nanostickers in spatial and temporal control of anchoring.

(2) The authors demonstrated the different “adhesion force” of the nanostickers using AFM. Can the authors provide more details on the AFM probe used? What is the surface mechanical and chemical properties of the probe? How is this set of measurement relevant to the bioadhesion between the hydrogel, the nanoparticle, and the tissue?

Response: No problem. We used the Bruker AFM probe (RTESPA-150) to measure the adhesion force, and the details of this AFM probe can be found at the Bruker official website (<https://www.brukerafmprobes.com/p-3911-rtespa-150.aspx>). As described on the official website, the high-sensitivity probe is composed of silicon, which may generate similar surface mechanical and chemical properties to silicon. By using this

commercial AFM probe, we could measure the adhesion force in a tapping mode. In detail, the AFM probe could accurately record the force and distance for its extension and retraction in the tested nanostickers-anchored tissues. As illustrated in Supplementary Fig. 1, the nanostickers serve as anchoring primers to bridge the tissues and attached hydrogels. The measured adhesion force could therefore reflect the anchorage of nanostickers on tissues, for which the strong anchoring ensured the tight connections of nanostickers on tissues, further contributing to the high adhesion energy of attached hydrogels.

(3) The authors demonstrated the application of the nanostickers in GI tract, which often involve various pH levels. How will the bioadhesion energy of the nanostickers be affected by the environmental pH? It has been shown that the pH can significantly affect the adhesion of chitosan-based bioadhesives.

Response: Thanks for the nice question. Although the pH of normal GI tracts in rats is changed between 6.0 and 8.0 (*Biochem. Biophys. Res. Commun.* **620**, 129-134 (2022)), the small pH range may not generate huge effects on the equilibrium between protonation and deprotonation of amine groups on chitosan, because the pH for abundant protonation of chitosan is usually below 5.5. Furthermore, the secure anchoring of nanostickers is primarily contributed by their high cohesive force due to the strong magnetic attraction. Therefore, the influence on the adhesion of nanostickers by the small pH range may be little.

(4) In the in vivo studies, the authors mentioned that a “levofloxacin-loaded (~200 mg) thin patch” was applied. What is the composition of such patch? Is it PAAm-alginate based hydrogel? What absorbable suture was used, since there’s no trace of suture remnant in the regions of interest shown in the histology images? What is the cause of death for animals that did not survive the surgeries (suture-only groups)?

Response: Thanks for the questions. This patch is composed of levofloxacin and PAAm-Alg hydrogel, which is prepared by directly immersing the PAAm-Alg hydrogel into levofloxacin solution. The absorbable suture was purchased from Shanghai Pudong Jinhuan Medical Supplies Co. Ltd. Because the histological section was very thin ~5 μm and performed after 10 days, which caused extensive cell migration and tissue integration (Supplementary Fig. 27), the slight suture might be covered by the regenerated tissues that could impede the observation. Due to the intense inflammation in the surgical sites, the suture-only groups suffered from serious ascites and tissue adhesion in the abdomen, and these complications are considered to be the main reasons for the death.

(5) The authors claimed that the “long-term integration with fragile tissues” was demonstrated, but only 10-day short-term implantation was conducted. Unless the author can demonstrate the biointegration of tissue-nanostickers for months, the reviewer suggests the authors to adjust their claims.

Response: Thanks for the rigorous advice. We have revised the description in the manuscript accordingly.

Reviewer #1 (Remarks to the Author):

The authors have provided satisfactory point-by-point answers. I therefore recommend publication of the manuscript as is.

Response: We sincerely thank the reviewer again for the highly positive comments on our manuscript.

Reviewer #2 (Remarks to the Author):

The authors have replied to my comments properly. I have only two remarks. The authors should check the consistency of the usage of the physical quantities related to magnetic field: the magnetic field strength H which unit is A/m and the magnetic flux density B which unit is T. The authors are recommended to check the consistency of the usage of the del (nabla) symbol for the magnetic force.

Response: We gratefully appreciate the reviewer again for the positive feedback and constructive suggestions. According to your suggestions, we have unified the usage of unit T for the magnetic flux density (B) and unit A/m for the magnetic field strength. The symbol (∇) for the magnetic force has been used in the whole manuscript.

Reviewer #3 (Remarks to the Author):

Response: We thank the reviewer for providing the valuable comments together with one of the reviewers.

Reviewer #4 (Remarks to the Author):

The reviewer's questions have been addressed. However, the methods section could be further improved by providing more details on the humane endpoints caused by the inflammation/complications.

Response: We gratefully thank the reviewer again for the positive comments to improve the quality of our manuscript. According to your suggestion, we have provided more details on the humane endpoints in the experimental section (page 20).

Reviewer #2 (Remarks to the Author):

It is still not clear how the authors work with the del operator, see the file attached. The authors may wish to consult the paper of T.H. Boyer, *Am. J. Phys.* 56, 688–692 (1988). Essentially, the authors believe that $\mathbf{F} = \nabla (\mathbf{m} \cdot \mathbf{B})$ and $\mathbf{F} = (\mathbf{m} \cdot \nabla) \mathbf{B}$ give the same the result if $\mathbf{m} = \text{const}$. However, a simple calculation shows that this is not the case:

$$\nabla(\mathbf{m} \cdot \mathbf{B}) = \nabla(m_x B_x + m_y B_y + m_z B_z) = \mathbf{e}_x \left(m_x \frac{\partial B_x}{\partial x} + m_y \frac{\partial B_y}{\partial x} + m_z \frac{\partial B_z}{\partial x} \right) + \dots$$

$$(\mathbf{m} \cdot \nabla) \mathbf{B} = \left(m_x \frac{\partial}{\partial x} + m_y \frac{\partial}{\partial y} + m_z \frac{\partial}{\partial z} \right) \mathbf{B} = \mathbf{e}_x \left(m_x \frac{\partial B_x}{\partial x} + m_y \frac{\partial B_x}{\partial y} + m_z \frac{\partial B_x}{\partial z} \right) + \dots$$

Since the authors rely on the identity of the two formulas, it is not obvious if their considerations are valid.

Response: We thank the reviewer's reminder and we are sorry for the confusion caused. Here we would like to describe in detail for the del operator of magnetic force. For the three-dimensional magnetic field generated by a sphere magnet, the magnetic flux density \mathbf{B} can be defined as $\mathbf{B}_{(x, y, z)}$. Compared to the huge changes of $\mathbf{B}_{(x, y, z)}$ at Z direction (Supplementary Fig. 16B and Fig. 16C) that generate large magnetic field gradients (*e.g.*, $\sim -39.4 \text{ mT cm}^{-1}$ at $d_{mt} = 30 \text{ mm}$), the $\mathbf{B}_{(x, y, z)}$ on the XY plane keeps almost the same (*e.g.*, $< 0.4 \text{ mT}$ of change at $d_{mt} = 30 \text{ mm}$) for X-coordinate and Y-coordinate from -25 mm to 25 mm (Fig. 3B and 3C), suggesting that the magnetic field gradients on the same XY plane (X and Y coordinates from -25 to 25) are negligible. As a result, the magnetic field gradients on one XY plane are dominated by z when the nanosticker (considered as a dot) is vertically above the center of magnet (original point), which means that the $\partial B_z / \partial z$ is remarkable, while the $\partial B_x / \partial x$, $\partial B_y / \partial x$, $\partial B_x / \partial y$, $\partial B_y / \partial y$, $\partial B_x / \partial z$, $\partial B_y / \partial z$, $\partial B_z / \partial x$, and $\partial B_z / \partial y$ are close to 0 because the change of magnetic flux density $\mathbf{B}_{(x, y, z)}$ on the same XY plane (X and Y coordinates from -25 to 25) is extremely small (close to 0). Therefore, $\mathbf{F} = \nabla (\mathbf{m} \cdot \mathbf{B})$ can be calculated as:

$$\mathbf{F} = \nabla (\mathbf{m} \cdot \mathbf{B}) = \mathbf{e}_x \left(m_x \frac{\partial B_x}{\partial x} + m_y \frac{\partial B_y}{\partial x} + m_z \frac{\partial B_z}{\partial x} \right) + \dots \approx \mathbf{e}_z \left(m_z \frac{\partial B_z}{\partial z} \right) \approx (\mathbf{m} \cdot \nabla) \mathbf{B}$$

In addition, we have consulted the suggested reference (*Am. J. Phys.* 56, 688-692 (1988)), which described that “ $\nabla (\mathbf{m} \cdot \mathbf{B}) = (\mathbf{m} \cdot \nabla) \mathbf{B} + \mathbf{m} \times (\nabla \times \mathbf{B})$. Comparing Eqs., it is clear that the two different force expressions for the two different magnetic dipole models agree if and only if $\mathbf{m} \times (\nabla \times \mathbf{B}) = 0$. For virtually all the examples in the textbooks $\nabla \times \mathbf{B} = 0$ and so no error is made in using the formulas interchangeably.” In accordance with this description, we continued to verify whether $\nabla \times \mathbf{B} = 0$:

$$\nabla \times \mathbf{B} = \begin{vmatrix} \mathbf{e}_x & \mathbf{e}_y & \mathbf{e}_z \\ \frac{\partial}{\partial x} & \frac{\partial}{\partial y} & \frac{\partial}{\partial z} \\ B_x & B_y & B_z \end{vmatrix} = \left(\frac{\partial B_z}{\partial y} - \frac{\partial B_y}{\partial z} \right) \mathbf{e}_x + \left(\frac{\partial B_x}{\partial z} - \frac{\partial B_z}{\partial x} \right) \mathbf{e}_y + \left(\frac{\partial B_y}{\partial x} - \frac{\partial B_x}{\partial y} \right) \mathbf{e}_z$$

Based on the calculation and former discussion, it also indicated that $\nabla \times \mathbf{B} = 0$

To make the calculation clear, we have revised the formula of magnetic force into $\mathbf{F} = \nabla (\mathbf{m} \cdot \mathbf{B})$ and provided the detailed calculation.

We have added “ $\mathbf{F}_{\text{gradient}} = \nabla (\mathbf{m} \cdot \mathbf{B}) = \mathbf{e}_x \left(m_x \frac{\partial B_x}{\partial x} + m_y \frac{\partial B_y}{\partial x} + m_z \frac{\partial B_z}{\partial x} \right) + \dots \approx \mathbf{e}_z \left(m_z \frac{\partial B_z}{\partial z} \right)$,

where the $\partial B_x / \partial x$, $\partial B_y / \partial x$, $\partial B_x / \partial y$, $\partial B_y / \partial y$, $\partial B_x / \partial z$, $\partial B_y / \partial z$, $\partial B_z / \partial x$, and $\partial B_z / \partial y$ are close to 0 because the magnetic flux density \mathbf{B} on the same XY plane (X and Y coordinates from -25 to 25) keeps almost the same (< 0.4 mT of change at $d_{mt} = 3$ cm)” and cited the important reference (*Am. J. Phys.* **56**, 688-692 (1988)) in the supporting information (page 10).

Reviewer #3 (Remarks to the Author):

Response: We thank the reviewer again for providing all the valuable comments.

Reviewer #4 (Remarks to the Author):

The reviewer's comments have been addressed. The acceptance of the manuscript is suggested.

Response: We thank the reviewer's suggestion on the acceptance of our manuscript.

Reviewer #2 (Remarks to the Author):

I want to retain from making a conclusion about the quality of the revision.

Response: In this revision, we have made a more in-depth calculation based on the given formula of magnetic force $\mathbf{F} = \nabla (\mathbf{m} \cdot \mathbf{B}) = (\mathbf{m} \cdot \nabla) \mathbf{B} + \mathbf{m} \times (\nabla \times \mathbf{B})$ in the paper (T. H. Boyer, Am. J. Phys., 56, 688-692 (1988)). We have carefully calculated all components of the magnetic flux density (\mathbf{B}) at different coordinates and provided the required values to calculate the $\nabla \times \mathbf{B}$ using COMSOL Multiphysics software, which has been widely used to do calculation and analysis for a three-dimensional magnetic field (e.g., Nat. Commun. 12, 3141 (2021); Nat. Commun. 14, 7493 (2023); Nat. Commun. 14, 2562 (2023)).

As shown in Fig. R1A and Fig. R1B, all components of \mathbf{B} along the X axis and Y axis at a certain Z-coordinate ($d_{mt} = 30$ mm) are calculated. All the components for Z-axis points at different d_{mt} are also calculated in Fig. R1C. As a typical example, when a nanosticker stands on a specific Z-axis point ($d_{mt} = 3$ cm), these components can be directly extracted as listed in Fig. R1D. Based on these results as you can observe, many components show a high rotational symmetry including $\partial B_y / \partial x$ is the same to $\partial B_x / \partial y$, $\partial B_z / \partial x$ is the same to $\partial B_x / \partial z$, and $\partial B_z / \partial y$ is the same to $\partial B_y / \partial z$, which should be able to fully address your question on the del operator. The high rotational symmetry in many components also indicates that the $\nabla \times \mathbf{B}$ is zero at that point. Then, we can get $\mathbf{F} = \nabla (\mathbf{m} \cdot \mathbf{B}) = (\mathbf{m} \cdot \nabla) \mathbf{B}$ for calculation.

Fig. R1. All components of the magnetic flux density along (A) X axis and (B) Y axis

at $d_{mt} = 3$ cm. (C) All components of the magnetic flux density for Z -axis points. (D) Concrete values for the components of magnetic flux density at a specific Z -axis point ($d_{mt} = 3$ cm). The components show a high rotational symmetry that indicate the $\nabla \times \mathbf{B}$ is zero at that point.

Reviewer #3 (Remarks to the Author):

Response: Thanks again for the reviewer's comment with the other reviewer.

Reviewer #5 (Remarks to the Author):

In the relation [Reference: T. H. Boyer, Am. J. Phys., 56, 688-692 (1988)]

$$\nabla (\mathbf{m} \cdot \mathbf{B}) = (\mathbf{m} \cdot \nabla) \mathbf{B} + \mathbf{m} \times (\nabla \times \mathbf{B}),$$

the second term on the right-hand side vanishes for irrotational fields since $\nabla \times \mathbf{B} = 0$ for such vector fields. This term is also zero when \mathbf{m} is parallel to $\text{curl}(\mathbf{B})$. The former (the curl-free constraint) is widely used in textbook examples as pointed out in the quoted/cited reference. In their revised manuscript the present authors indicate that the variations of magnetic flux density gradients are negligibly small in the plane perpendicular to the z -axis. Consequently, they arrive at $\nabla \times \mathbf{B} \approx 0$ justifying the equivalence of $\nabla (\mathbf{m} \cdot \mathbf{B})$ and $(\mathbf{m} \cdot \nabla) \mathbf{B}$ (approximately) in their calculation of the magnetic force. Given the magnetic field gradients range in the XY plane, this is probably an acceptable reasoning. But the validity of this sort of justification in a truly three-dimensional case is questionable!

Response: Your comments are logical and meticulous. For a detailed validation process, we have calculated all components of the magnetic flux density (\mathbf{B}) at different coordinates (Fig. R2A, Fig. R2B, and Fig. R2C) and provided the required values to calculate the $\nabla \times \mathbf{B}$ in the three-dimensional magnetic field. All components of \mathbf{B} at different coordinates can be directly extracted from these figures. As a representative example, the components for a nanosticker standing on a specific Z -axis point ($d_{mr} = 3$ cm) are listed in Fig. R2D, showing a high rotational symmetry including $\partial B_y / \partial x$ is the same to $\partial B_x / \partial y$, $\partial B_z / \partial x$ is the same to $\partial B_x / \partial z$, and $\partial B_z / \partial y$ is the same to $\partial B_y / \partial z$, which makes the $\nabla \times \mathbf{B}$ equal to zero at that point. As you may know, it is impossible to measure these components by experimental instruments especially for a nanoscale dot (nanosticker) around 10 nm. In this premise, we have to simulate the three-dimensional magnetic field to calculate these components using COMSOL Multiphysics software, which has been widely used to calculate and analyze a three-dimensional magnetic field in many excellent papers (e.g., *Nat. Commun.* **12**, 3141 (2021); *Nat. Commun.* **14**, 7493 (2023); *Nat. Commun.* **14**, 2562 (2023)).

Fig. R2. All components of the magnetic flux density along (A) X axis and (B) Y axis at $d_{mt} = 3 \text{ cm}$. (C) All components of the magnetic flux density for Z-axis points. (D) Concrete values for the components of magnetic flux density at a specific Z-axis point ($d_{mt} = 3 \text{ cm}$). The components show a high rotational symmetry that indicate the $\nabla \times \mathbf{B}$ is zero at that point.

Indeed, compared to the ideal environment in numerical simulations, there may exist some differences for a real-world spherical permanent magnet such as the non-uniform magnetization during factory process. However, as a numerical simulation, it should be inevitable to require some preconditions including the assumption of uniform magnetization, because the results in numerical simulations can only be obtained with some specific assumptions or approximations. Therefore, the question actually becomes whether the numerical simulations could be used to well reflect the truly three-dimensional magnetic field.

In a real condition, the nanostickers were rotated around Z axis by a spherical magnet (magnetization direction along the XY plane) at a low rotational frequency of 3 Hz (3 revolutions per second), and thus $\nabla \times \mathbf{B}$ could be described by one of Maxwell's equations (Ampère-Maxwell Law): $\nabla \times \mathbf{B} = \mu_0 (\mathbf{J} + \epsilon_0 \frac{\partial \mathbf{E}}{\partial t})$. In this equation, μ_0 (vacuum permeability) and ϵ_0 (vacuum permittivity) are constants, \mathbf{J} represents the current density, and \mathbf{E} represents the electric field intensity, which are two variables in a certain environment. \mathbf{J} can be decomposed into free current density (\mathbf{J}_{free}) and bound current

density ($\mathbf{J}_{\text{bound}}$), and $\mathbf{J}_{\text{bound}} = \nabla \times \mathbf{M}$ (\mathbf{M} represents the magnetization of materials), making $\nabla \times \mathbf{B} = \mu_0 (\mathbf{J}_{\text{free}} + \nabla \times \mathbf{M} + \epsilon_0 \frac{\partial \mathbf{E}}{\partial t})$. From this perspective, the first term \mathbf{J}_{free} is 0 because free current is existed in macroscopic conductors with external electric power, but the nanostickers have poor electrical conductivity (coated with insulating chitosan) and there was no supplied electric power to the nanostickers; the second term $\nabla \times \mathbf{M}$ is also 0 because \mathbf{M} is regarded as a constant vector based on the fact that the nanosticker around 10 nm is quite small compared to the spatial variation of the external magnetic field; the third term $\epsilon_0 \frac{\partial \mathbf{E}}{\partial t}$ equals to 0 or extremely close to 0 due to the poor electrical conductivity of nanostickers (coated with insulating chitosan), low rotational frequency (3 revolutions per second) of magnet, and quite small volume of nanostickers (around 10 nm) compared to the magnetic field in space, which are all enormous barriers to the generation of eddy current. In short, we think that $\nabla \times \mathbf{B}$ equals or extremely approaches to 0 is rational in our experimental settings in terms of the above three terms, and therefore we can consider that the formula of magnetic force $\nabla (\mathbf{m} \cdot \mathbf{B})$ is also equivalent or extremely close to $(\mathbf{m} \cdot \nabla) \mathbf{B}$ in a real condition.

Overall, we elucidate the calculation and analysis process as far as we can from several aspects, and all of them point out $\nabla \times \mathbf{B}$ equals or extremely approaches to 0. We agree that numerical simulations are based on assumptions or approximations and thus do not represent exactly what happens in the real world. However, the simulations are still universal tools that can demonstrate the physical process, especially in cases where experiments are difficult to carry out. We also elaborate the real conditions based on our experimental settings to provide the detailed analysis for all factors step by step. We hope the in-depth calculation and analysis for the three-dimensional magnetic field can help you consider that our justification is now sufficient and valid.

Reviewer #5 (Remarks to the Author):

The authors provide additional justification for the 'almost equivalence' of the two expressions $\nabla (\mathbf{m} \cdot \mathbf{B})$ and $(\mathbf{m} \cdot \nabla) \mathbf{B}$ in the calculation of magnetic force. **Fig R2** depicts the calculated values indicating that $\nabla \times \mathbf{B} \approx 0$, showing rotational symmetry in their set-up. Another reasoning is supplied via Ampere's law using nanostickers setting considered in their work, which leads to the (approximate) irrotationality of the magnetic flux density. This latter (related) approach is also used in the reference [T. H. Boyer, *Am. J. Phys.*, **56**, 688-692 (1988)] to illustrate the non-equivalence of the two force expressions in other scenarios.

I agree that many realistic three-dimensional magnetic fields might not be captured using COMSOL Multiphysics software, and that pre-conditioning is required for numerical simulations. The authors calculations and analysis offer a theoretical justification that is comparatively better in the area of their study.

Response: We thank the reviewer's positive attitude towards our additional justification for the magnetic force.